# Learning Procedural Abstractions and Evaluating Discrete Latent Temporal Structure

**Karan Goel**
Department of Computer Science
Stanford University
`kgoel@cs.stanford.edu`

**Emma Brunskill**
Department of Computer Science
Stanford University
`ebrun@cs.stanford.edu`

## Abstract

Clustering methods and latent variable models are often used as tools for pattern mining and discovery of latent structure in time-series data. In this work, we consider the problem of learning procedural abstractions from possibly high-dimensional observational sequences, such as video demonstrations. Given a dataset of time-series, the goal is to identify the latent sequence of steps common to them and label each time-series with the temporal extent of these procedural steps. We introduce a hierarchical Bayesian model called PRISM that models the realization of a common procedure across multiple time-series, and can recover procedural abstractions without supervision. We also bring to light two characteristics ignored by traditional evaluation criteria when evaluating latent temporal labelings (*temporal clusterings*) – *segment structure*, and *repeated structure* – and develop new metrics tailored to their evaluation. We demonstrate that our metrics improve interpretability and ease of analysis for evaluation on benchmark time-series datasets. Results on benchmark and video datasets indicate that PRISM outperforms standard sequence models as well as state-of-the-art techniques in identifying procedural abstractions.

## 1 Introduction

A fundamental problem in machine learning is the discovery of structure in unsupervised data. In this work we are particularly interested in uncovering the latent structure in procedural data – potentially high-dimensional observational time-series resulting from some latent sequence of events. For example, a video showing how to change a tire might involve jacking up the car, removing one nut from the wheel, removing a second nut, etc. There exists an enormous wealth of videos of procedures available and inferring the latent structure of such videos could be useful in a huge range of applications, from supporting search queries ("find all segments where someone jacks up the car") to adding to nascent work on learning from observation (Borsa et al., 2017; Stadie et al., 2017; Torabi et al., 2018) in which robots may be taught merely by observing a procedure performed in a video. Procedural learning is related to activity recognition and the broader field of latent temporal dynamical learning, but focuses on the simpler but still ubiquitous setting where the latent activity sequence is a fixed procedure.

As we started to develop methods for performing latent procedure inference from temporal data, we considered how to evaluate our resulting methods. One important evaluation protocol for unsupervised learning methods is *external evaluation*, where predicted labelings of the discrete variables are compared to ground-truth labels (Rosenberg & Hirschberg, 2007). But the precise criteria for this evaluation is critical. Though there are an enormous number of methods for modeling and inferring latent structure in dynamical systems, including Hidden Markov models, dynamic Bayesian networks, and temporal clustering for time-series data *e.g.* Fox et al. (2008b; 2014); Krishnan et al. (2015); Linderman et al. (2017); Zhou et al. (2008; 2013); Vidal & Favaro (2014), the external evaluation for such approaches is typically done using the same clustering metrics that are used for non-temporal data: metrics like the normalized mutual information (NMI) (Strehl & Ghosh, 2002) or by computing cluster correspondences using the Munkres algorithm (Munkres, 1957). Unfortunately such metrics disregard temporal information and can therefore make it hard to assess important considerations that arise in latent temporal structure extraction, including in procedural inference.

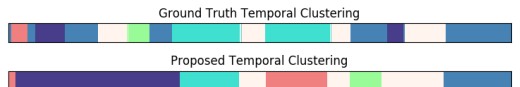

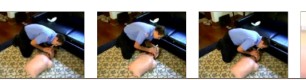

(a) Prediction by PRISM for a *cpr* video. Note how repeated structure (cream primitive) is recovered.

(b) Repeated structure captured by PRISM at different points in the video (3 left images) and across videos (right image).

Figure 1: Procedural learning. Given an input set of temporal sequences all demonstrating the same procedure we wish to infer the latent structure of that procedure. Here we show the results of our method, PRISM, on inferring a *cpr* procedure from a set of videos in the INRIA Instructional videos dataset. The ground truth temporal clustering on the upper left represents a human labeling of the activities involved in a particular *cpr* video where color denotes the same activity. For example, the activity denoted by the cream color occurred 4 times across the sequence.

To address the limitations of prior evaluation criteria for the external evaluation of discrete latent temporal structure, our first contribution is the creation of new criteria. We identify at least two key aspects of temporal latent structure discovery that are not captured well with existing non-temporal metrics: *segment structure*, the temporal durations between transitions from one discrete latent state to another, and *repeated structure*, where a particular latent variable may appear in multiple segments. Figure1 shows an example latent sequence associated with a person performing cardiopulmonary resuscitation (*cpr*) – the colors denote the activity type (such as blowing air into the lungs) and the temporal duration of a fixed activity type is shown as a continuous block of color. Our new metrics can evaluate both segment structure and repeated structure, and we later demonstrate how these metrics may be of broader interest in latent temporal dynamical learning, as they can illustrate and illuminate different strengths and weaknesses of existing work on common benchmark datasets.

Equipped with new measures for evaluating success, we return to our key goal, inferring the latent procedure observed in a set of high dimensional demonstrations. Past expressive models like Hidden Markov models can struggle in this setting, since the resulting procedure may not be Markov: while the setting can be forced to appear Markov by employing the history as part of the state, this can reduce sample efficiency (by artificially increasing the latent space size). Additionally assuming the system is Markov when it is not can mean inferring stochastic latent dynamical structure when it is in fact deterministic. Recent work (Sener & Yao, 2018) tackles a closely related setting, but does not allow repeated primitives, which is a key aspect of many procedures. Instead we introduce a new generative Bayesian model (PRISM) for identifying procedural abstractions without supervision. Results on datasets for procedural tasks – including surgical demonstrations and how-to videos – show that our method can learn procedural abstractions effectively compared with state-of-the-art methods and popular baselines, and outperform prior work on both our new metrics and classic metrics when repeated structure is present.

## 2 DEFINITIONS AND SETTING

**Temporal Clustering.** A *time-series* $\mathbf{X}^i = (\mathbf{x}_1^i, \mathbf{x}_2^i, \ldots, \mathbf{x}_{m_i}^i)$, $i \in [n]$ is a sequence of feature vectors (or items) of length $m_i$ with $\mathbf{x}_t^i \in \mathbb{R}^d$. A *temporal clustering* $\mathcal{C} = (c_1, \ldots, c_m)$ is a sequence of cluster labels with $c_t \in \Gamma_\mathcal{C}$ where $\Gamma_\mathcal{C}$ is a set of cluster labels. Temporal clusterings map each item in the feature trajectory $\mathbf{X}$ to a cluster label in $\Gamma_\mathcal{C}$. We use $\mathcal{C}^i$ as shorthand to denote the temporal clustering generated by some method for $\mathbf{X}^i$. The ground truth temporal clustering for $\mathbf{X}^i$ will be denoted by $\mathcal{G}^i$, with corresponding ground truth label set $\Gamma_\mathcal{G}$. For simplicity, we define $\mathcal{C}_{a:b} = (c_a, c_{a+1}, \ldots, c_b)$ where $1 \leq a \leq b \leq m$ are time indices. For $\alpha \in \Gamma_\mathcal{C}$, let $\mathcal{C}[\alpha] = \{t | c_t = \alpha, 1 \leq t \leq m\}$ be the set of time indices in $\mathcal{C}$ whose cluster labels equal $\alpha$. $\mathcal{C}[\alpha]$ is the cluster or partition (in the standard sense) corresponding to the label $\alpha$. Temporal clusterings are distinct from time-series segmentation (Chung et al., 2004) and changepoint detection (Killick et al., 2012), which consider only identifying boundaries of different dynamic regimes in the time-series.

**Segment.** A *segment* is a contiguous sequence of identical cluster labels in $\mathcal{C}$. Each cluster in $\mathcal{C}$ may be split across multiple segments. We represent each segment as a pair of time indices $(a, b)$, where $a$ represents the start time-index of the segment while $b$ the end time-index (both included). Formally, we let $\mathcal{S}(\mathcal{C}, \alpha) = \{(a, b) | \mathcal{C}_{a:b} = (c_a = \alpha, c_{a+1} = \alpha, \ldots, c_b = \alpha), 1 \leq a \leq b \leq m\}$ be a function

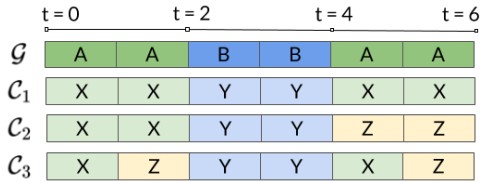

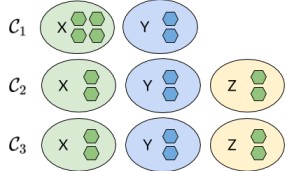

(a) $\mathcal{G}$ is the ground truth temporal clustering and $\mathcal{C}_1, \mathcal{C}_2, \mathcal{C}_3$ are possible temporal clusterings.

(b) $\mathcal{C}_1, \mathcal{C}_2, \mathcal{C}_3$ when converted to standard non-temporal clusterings for use with traditional metrics.

Figure 3: Problematic cases for traditional metrics, on a simple 6-step time-series.

that maps a temporal clustering $\mathcal{C}$ and cluster label $\alpha \in \Gamma$ to the set of segments associated with $\alpha$ in $\mathcal{C}$. We let $\mathfrak{S}_\mathcal{C} = \bigcup_{\alpha \in \Gamma_\mathcal{C}} \mathcal{S}(\mathcal{C}, \alpha)$ be the set of all segments in $\mathcal{C}$.

**Procedure.** Let $\mathcal{P}$ be a function that removes running duplicates from a sequence of labels, yielding a sequence of *tokens*. For instance, $\mathcal{P}(A, A, A, B, B, B, C, C, C, A, A, B) = (A, B, C, A, B)$ yielding 5 tokens. For a temporal clustering $\mathcal{C}$, we define its corresponding *procedure* to be $\mathcal{P}(\mathcal{C})$. The procedure captures the primitives (as tokens) present in the temporal clustering and the sequence in which they occur. We say that $\mathcal{C}$ exhibits *repeated structure* if atleast one primitive is reused in $\mathcal{C}$. This corresponds to having multiple tokens of the same cluster label in $\mathcal{P}(\mathcal{C})$. The procedure $(A, B, C, A, B)$ contains repeated structure since both A and B are reused. We also define a weight function $\mathcal{W}$ that counts the number of running duplicates in a sequence *e.g.* $\mathcal{W}(A, A, A, B, B, B, C, C, C, A, A, B) = (3, 3, 3, 2, 1)$. Notice that $\mathcal{W}$ computes the length of each segment in the sequence. Weights in $\mathcal{W}(\mathcal{C})$ thus have a one-to-one correspondence with tokens in $\mathcal{P}(\mathcal{C})$,

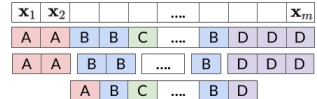

Figure 2: From top – time-series, temporal clustering, segments and procedure.

equaling the length of each token in the procedure. Taken together $\mathcal{W}(\mathcal{C}), \mathcal{P}(\mathcal{C})$ can be used to recover $\mathcal{C}$. Lastly, we let $\mathcal{H}(\mathcal{P}_1, \mathcal{P}_2, \mathcal{W}_1, \mathcal{W}_2)$ be a similarity measure between 2 temporal clusterings represented by $\mathcal{P}_1, \mathcal{W}_1$ and $\mathcal{P}_2, \mathcal{W}_2$ respectively.

Fig 2 illustrates these ideas for a temporal clustering defined over the label set $\Gamma = \{A, B, C, D\}$. The procedure exhibits repeated structure since B has (atleast) 2 tokens/segments corresponding to it.

## 3  EVALUATING LATENT TEMPORAL STRUCTURE

External clustering evaluation is the process of comparing a clustering to a ground-truth labeling in order to evaluate it. Unfortunately, as we discuss below, using traditional clustering evaluation criteria is problematic for evaluating temporal clusterings.

**Existing evaluation criteria.** The use of standard clustering metrics for evaluation is prevalent in prior work that attempts to evaluate latent temporal structure (Zhou et al., 2008; 2013; Fox et al., 2008b; 2009; 2014; Krishnan et al., 2015; Sener & Yao, 2018; Hoai & De la Torre, 2012). For a temporal clustering $\mathcal{C}$ the widely used *purity* (Rosenberg & Hirschberg, 2007) metric penalizes the presence of items from different ground truth labels in the same cluster. This is a desirable property that a clustering evaluation criterion should have. Purity is defined as purity $= \frac{1}{m} \sum_{\alpha \in \Gamma_\mathcal{C}} \max_{\beta \in \Gamma_\mathcal{G}} |\mathcal{G}[\beta] \cap \mathcal{C}[\alpha]|$, A related metric is homogeneity (Rosenberg & Hirschberg, 2007), which captures a similar idea in a different way: homogeneity $= 1 - \frac{H(\Gamma_\mathcal{G}|\Gamma_\mathcal{C})}{H(\Gamma_\mathcal{G})}$ where $H(\Gamma_\mathcal{G})$ is the entropy of the clustering $\mathcal{G}$ when partitioned by the labels $\Gamma_\mathcal{G}$, $H(\Gamma_\mathcal{G}|\Gamma_\mathcal{C})$ is the conditional entropy of $\mathcal{G}$ given $\mathcal{C}$. Intuitively the conditional entropy term looks inside every cluster in $\mathcal{C}$, and checks the entropy of the items inside in terms of the ground truth labels. The conditional entropy will be low (and homogeneity high) if every cluster in $\mathcal{C}$ contains items of only a single ground truth label. Another important metric is *completeness* (Rosenberg & Hirschberg, 2007), which prefers clusterings where all items from a ground-truth label lie in the same cluster *i.e.* the ground truth label is not split across clusters. Completeness is defined identically to homogeneity, but with the clusterings swapped $\mathcal{G} \longleftrightarrow \mathcal{C}$.

Both completeness and homogeneity are considered important criteria for clustering. Maximizing either at the expense of the other is bad – high homogeneity and low completeness imply a clustering that is too fine-grained (*e.g.* every item in its own cluster), while low homogeneity and high completeness imply a clustering that is too coarse (*e.g.* a single cluster containing all items). Therefore, metrics that balance these 2 criterion often perform best (Rosenberg & Hirschberg, 2007) – *normalized mutual information* (NMI) (Strehl & Ghosh, 2002) and *V-measure* (Rosenberg & Hirschberg, 2007) are examples of such metrics.

Another popular suite of metrics uses the Munkres algorithm (Zhou et al., 2008) – a one-to-one correspondence between clusters and ground-truth labels is computed so that any classification criterion (commonly accuracy) under this correspondence is maximized.

**Characterizing latent temporal structure.** Two major characteristics distinguish the evaluation of temporal clusterings from the standard clustering case.

- *Repeated structure.* The presence of repeated structure in ground-truth – ground-truth labels that are split across several segments in the trajectory. Ideally, we want that this repeated structure is identified correctly *e.g.* in Fig 3 identifying that the segments $A_{0:2}$, $A_{4:6}$ are repetitions of the same primitive.
- *Segment structure.* The locations and sizes of the segments in ground-truth *e.g.* in Fig 3 we would like temporal clusterings that switch between primitives at exactly the time points $t = 2, 4$ with the same 3 segment structure.

Neither repeated or segment structure are seen in standard clusterings, since they treat data as a bag of items.

**Limitations of existing criteria.** Existing metrics cannot score repeated or segment structure because they convert temporal clusterings to standard clusterings – a *contingency matrix* (Rosenberg & Hirschberg, 2007) – before evaluation. This erases the temporal information important to score these ideas. In fact, qualitatively distinct temporal clusterings can be mapped to the same contingency matrix, and receive the same score, which is clearly undesirable.

We use Fig 3 as an illustrative example to highlight these shortcomings. In Fig 3, $\mathcal{C}_1$ perfectly matches $\mathcal{G}$, while both $\mathcal{C}_2$ and $\mathcal{C}_3$ split the cluster $\mathcal{G}[A]$ – $\mathcal{C}_2$ into $\mathcal{C}_2[X]$ and $\mathcal{C}_2[Z]$ and $\mathcal{C}_3$ into $\mathcal{C}_3[X]$ and $\mathcal{C}_3[Z]$. In addition, $\mathcal{C}_2$ *also* misses the repeated structure in A while $\mathcal{C}_3$ does not. $\mathcal{C}_3$ captures this repeated structure as XZ, with $X_{0:1}Z_{1:2}$ corresponding to $A_{0:2}$ and $X_{4:5}Z_{5:6}$ corresponding to $A_{4:6}$. However, it does so at the cost of creating more segments than $\mathcal{C}_2$ – ignoring labels, $\mathcal{C}_2$ has a segmentation exactly like $\mathcal{G}$. Intuitively, we prefer $\mathcal{C}_1 > \mathcal{C}_3 > \mathcal{C}_2$ if we care more about repeated structure, and $\mathcal{C}_1 > \mathcal{C}_2 > \mathcal{C}_3$, if we care more about the segmentation quality.

However, all 3 temporal clusterings $\mathcal{C}_1, \mathcal{C}_2, \mathcal{C}_3$ in Fig 3 are assigned a perfect purity or homogeneity, since as seen in Fig 3b, no cluster in any of the 3 contains items from more than one ground-truth label. Moreover, *no* traditional clustering criteria can distinguish $\mathcal{C}_2$ from $\mathcal{C}_3$ since their cluster composition is identical, as seen in Fig 3b. We now discuss new evaluation criteria that can systematically evaluate and score both repeated and segment structure.

## 3.1 Evaluating Repeated Structure

We describe a new evaluation criteria, called the *repeated structure score (RSS)* below. Algorithmically, the calculation of the RSS requires the following steps,

1. Pick a ground-truth label $\beta \in \Gamma_{\mathcal{G}}$ and find the set of segments $\mathcal{S}(\mathcal{G}, \beta)$ marked with $\beta$ in $\mathcal{G}$. These segments all exhibit repeated structure in $\mathcal{G}$.

2. Find $\mathcal{C}_{a:b}$ for every $(a, b) \in \mathcal{S}(\mathcal{G}, \beta)$. This is the predicted temporal clustering restricted to a particular segment $(a, b)$ in $\mathcal{G}$.

3. For every such pair $\mathcal{C}_{a_1:b_1}, \mathcal{C}_{a_2:b_2}$, use a scoring function $\mathcal{H}$ to compute $\mathcal{H}(\mathcal{C}_{a_1:b_1}, \mathcal{C}_{a_2:b_2})$.

4. Aggregate these scores across all such pairs for every ground-truth label $\beta \in \Gamma_{\mathcal{G}}$,

$$\text{RSS} = \frac{\sum_{\beta \in \Gamma_{\mathcal{G}}} \sum_{(a_1,b_1),(a_2,b_2) \in \mathcal{S}(\mathcal{G},\beta)} \mathcal{H}(\mathcal{C}_{a_1:b_1}, \mathcal{C}_{a_2:b_2})}{2 \sum_{\beta \in \Gamma_{\mathcal{G}}} |\mathcal{S}(\mathcal{G},\beta)| \sum_{(a,b) \in \mathcal{S}(\mathcal{G},\beta)} (b - a + 1)}$$

where the denominator normalizes RSS to lie in $[0, 1]$.

As a concrete example, in Fig 4a, we could pick A in $\mathcal{G}$ for step 1 with $\mathcal{S}(\mathcal{G}, \beta = A) = \{(0, 2), (4, 7)\}$. For step 2, we find $\mathcal{C}_{0:2} = (X, X)$ and $\mathcal{C}_{4:7} = (X, Z, Z)$. For step 3, we must choose an appropriate scoring function $\mathcal{H}$. While many choices for $\mathcal{H}$ are possible, we let $\mathcal{H}(\mathcal{C}_{a_1:b_1}, \mathcal{C}_{a_2:b_2})$ be the total weight of the *heaviest common sub-sequence* or *substring* (Jacobson & Vo, 1992) in $\mathcal{C}_{a_1:b_1}$ and $\mathcal{C}_{a_2:b_2}$. Fig 4b uses this choice of $\mathcal{H}$ to score the pair of segments we picked – we run $\mathcal{H}((X, X), (X, Z, Z))$ and it returns a score of 3.

Intuitively, $\mathcal{H}$ tries to match procedures $\mathcal{P}(\mathcal{C}_{a_1:b_1})$ to $\mathcal{P}(\mathcal{C}_{a_2:b_2})$ – finding the (heaviest) sequence of tokens common to both $\mathcal{P}(\mathcal{C}_{a_1:b_1})$ and $\mathcal{P}(\mathcal{C}_{a_2:b_2})$, while respecting the temporal ordering of both procedures. At one extreme, if $\mathcal{H}$ evaluates to 0 then *no* overlapping tokens exist in $\mathcal{P}(\mathcal{C}_{a_1:b_1})$ and $\mathcal{P}(\mathcal{C}_{a_2:b_2})$. This implies that $\mathcal{C}$ failed to identify repeated structure in the two segments and a score of 0 is appropriate. At the other extreme, if both segments follow identical procedures $\mathcal{P}(\mathcal{C}_{a_1:b_1}) = \mathcal{P}(\mathcal{C}_{a_2:b_2})$, they receive the maximum possible score according to $\mathcal{H}$; reflecting that $\mathcal{C}$ identified repeated structure perfectly. $\mathcal{H}$ can be computed efficiently using a dynamic program in $O(|\mathcal{P}_1||\mathcal{P}_2|)$.

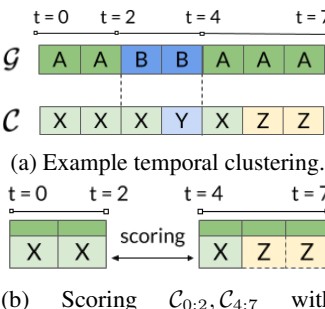

(a) Example temporal clustering.

(b) Scoring $\mathcal{C}_{0:2}, \mathcal{C}_{4:7}$ with $\mathcal{P}(\mathcal{C}_{0:2}) = (X)$, $\mathcal{W}(\mathcal{C}_{0:2}) = (2)$, $\mathcal{P}(\mathcal{C}_{4:7}) = (X, Z)$, $\mathcal{W}(\mathcal{C}_{4:7}) = (1, 2)$ and running $\mathcal{H}$ yields heaviest sub-sequence $(X)$ with score 3. $\mathcal{C}_{5:7}$ (dotted) does not contribute to the score.

Figure 4: Example for repeated structure scoring.

## 3.2 Evaluating Segment Structure

We first introduce a new information-theoretic criteria – the *label-agnostic segmentation score (LASS)*. The main purpose of LASS is to score how well $\mathcal{C}$ predicts the location of transition points between segments in $\mathcal{G}$. We want to identify and score cases where $\mathcal{C}$ either oversegments (introduces extra segments) or undersegments (omits segments) the time-series compared to $\mathcal{G}$. In any temporal clustering, the transition boundaries do not depend on the actual labeling, but only on where the labels switch. Therefore, LASS is label-agnostic and only requires the transition points between segments.

Prior work in changepoint detection and time-series segmentation (Killick et al., 2012) has attempted to quantify this using detection criteria – *e.g.* if a changepoint close to one in ground-truth is detected, a true positive is recorded. However, these criteria (i) require specifying the tolerance interval for detection, and (ii) don't degrade gracefully with the segmentation, which limits their applicability to temporal clusterings.

Recall that $\mathfrak{S}_\mathcal{G}, \mathfrak{S}_\mathcal{C}$ are the set of all segments in $\mathcal{G}$ and $\mathcal{C}$. To compute LASS, the main quantity we rely on is $H(\mathfrak{S}_\mathcal{C}|\mathfrak{S}_\mathcal{G})$ – a conditional entropy over segments in $\mathcal{C}$. This can be written as,

$$H(\mathfrak{S}_\mathcal{C}|\mathfrak{S}_\mathcal{G}) = -\sum_{(a,b)\in\mathfrak{S}_\mathcal{G}} \frac{(b-a+1)}{n} \sum_{(\tilde{a},\tilde{b})\in\mathfrak{S}_{\mathcal{C}_{a:b}}} \frac{(\tilde{b}-\tilde{a}+1)}{(b-a+1)} \log \frac{(\tilde{b}-\tilde{a}+1)}{(b-a+1)}$$

Intuitively, for each ground-truth segment $(a, b) \in \mathfrak{S}_\mathcal{G}$, we compute the (weighted) entropy of $\mathcal{C}$ restricted to $(a, b)$. This entropy will be $> 0$ only if $\mathcal{C}$ creates new segments in $(a, b)$. On the other hand, to detect under-segmentation, we can simply invert the roles of $\mathcal{G}$ and $\mathcal{C}$ by calculating $H(\mathfrak{S}_\mathcal{G}|\mathfrak{S}_\mathcal{C})$. Finally, noting that $H(\mathfrak{S}_\mathcal{C}|\mathfrak{S}_\mathcal{G}) \le H(\mathfrak{S}_\mathcal{C})$, we define LASS $:= 1 - \frac{H(\mathfrak{S}_\mathcal{C}|\mathfrak{S}_\mathcal{G})+H(\mathfrak{S}_\mathcal{G}|\mathfrak{S}_\mathcal{C})}{H(\mathfrak{S}_\mathcal{C})+H(\mathfrak{S}_\mathcal{G})}$. Over-segmentation (and under-segmentation) can also be measured separately by defining LASS$-$O $:= 1 - \frac{H(\mathfrak{S}_\mathcal{C}|\mathfrak{S}_\mathcal{G})}{H(\mathfrak{S}_\mathcal{C})}$ and LASS$-$U $:= 1 - \frac{H(\mathfrak{S}_\mathcal{G}|\mathfrak{S}_\mathcal{C})}{H(\mathfrak{S}_\mathcal{G})}$. The algebraic form chosen for the criteria relates it back to existing information-theoretic criteria such as the NMI (Vinh et al., 2010).

Next, we extend the completeness metric defined earlier. While completeness is desirable for any temporal clustering, using completeness directly has an unintended drawback – it penalizes repeated structure. An example of this can be seen in Fig 3a, where $\mathcal{C}_3$ is assigned the same completeness score as $\mathcal{C}_2$ despite capturing repeated structure. Since the RSS already provides a systematic way of evaluating presence/absence of repeated structure, we modify the completeness score slightly. We define *segmental completeness* to be $1 - \frac{H(\Gamma_\mathcal{C}|\mathfrak{S}_\mathcal{G})}{H(\Gamma_\mathcal{C})}$, where the conditioning is now on ground-truth

segments rather than on ground-truth labels. Equivalently, we can view this as relabeling each segment in ground-truth to be distinct, and then computing completeness. In this way, segmental completeness serves a similar function to completeness without interfering with repeated structure scoring. Homogeneity can also be extended to define *segmental homogeneity* similarly.

Lastly, we define an aggregate *segment structure score (SSS)*,

$$\text{SSS} := 1 - \frac{H(\mathfrak{S}_{\mathcal{C}}|\mathfrak{S}_{\mathcal{G}}) + H(\mathfrak{S}_{\mathcal{G}}|\mathfrak{S}_{\mathcal{C}}) + H(\Gamma_{\mathcal{C}}|\mathfrak{S}_{\mathcal{G}}) + H(\Gamma_{\mathcal{G}}|\mathfrak{S}_{\mathcal{C}})}{H(\mathfrak{S}_{\mathcal{C}}) + H(\mathfrak{S}_{\mathcal{G}}) + H(\Gamma_{\mathcal{C}}) + H(\Gamma_{\mathcal{G}})}$$

### 3.3 A Combined Evaluation Criteria

We have introduced several new metrics: (i) the RSS systematically scores repeated structure; (ii) the LASS scores how well the transition structure is captured; (iii) the SSS provides a unified metric for assessing segment structure. Note that all of these metrics obey the useful property of $n$-invariance (Rosenberg & Hirschberg, 2007; Meilă, 2007) – scaling the number of items (stretching the temporal clustering) does not change the metric values. Finally, we provide a single evaluation measure for temporal clusterings, the *temporal structure score (TSS)* that balances the RSS with the SSS.

$$\text{TSS} := \frac{(1 + \beta) \cdot \text{RSS} \cdot \text{SSS}}{\beta \cdot \text{RSS} + \text{SSS}}$$

While it is preferable to examine the constituent scores individually to determine the strengths and weaknesses of methods, a unified metric allows us to choose methods that balance these criteria. $\beta$ can be tuned to change the influence of the constituent metrics according to the desired application – we set $\beta = 1.0$ in a problem-agnostic fashion in this paper. This choice is in line with previous work that defines compound criteria, such as Rosenberg & Hirschberg (2007).

## 4 Identifying Procedural Abstractions

So far, we have argued the importance of evaluating repeated and segment structure correctly in temporal data. Our new metrics enable a systematic evaluation of these ideas, whereas existing metrics did not specifically evaluate or emphasize them. Given this, we now return to our original goal of identifying procedural abstractions in time-series data, which we term *procedure identification*.

Concretely, we assume a dataset of time-series such that each demonstrates the same task or phenomenon. Crucially, we assume that all time-series share this single, latent procedure – different time-series are distinct realizations of this underlying procedure. The goal is to label the temporal structure present in each time-series while identifying the common, latent procedure. For example, this may involve identifying the common sequence of steps in a set of demonstrations that follow the same cooking recipe as well as their temporal extent in each demonstration.

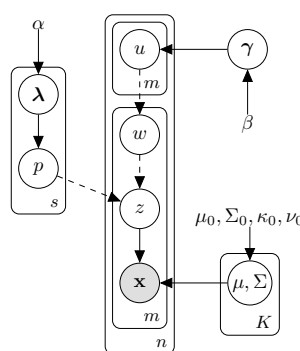

Figure 5: PRISM model

Procedure identification is related to recent efforts in computer vision that relate to unsupervised activity recognition in videos (Wu et al., 2015; Yang et al., 2013; Krishnan et al., 2015; Sener & Yao, 2018) as well as prior work in latent variable modeling (Yang et al., 2013; Fox et al., 2008b; 2014; Johnson & Willsky, 2013; Linderman et al., 2017). These papers typically assume a general setting where activities or latent tokens may be reused within or across multiple time-series but there is no onus on methods to discover this structure. Unlike prior work, procedure identification requires that methods also return a common procedure in addition to the temporal clustering of each time-series. This additional requirement places a burden on the method to *explicitly*, and not implicitly reason about repeated structure.

Formally, finding a common procedure $\mathcal{P}$ corresponds to finding temporal clusterings $\mathcal{C}^i$ for each time-series, such that $\mathcal{P} = \mathcal{P}(\mathcal{C}^1) = \mathcal{P}(\mathcal{C}^2) = \cdots = \mathcal{P}(\mathcal{C}^n)$. Since a single procedure consistent with all temporal clusterings is desired, shared primitives *must* be identified across all videos. This requirement also makes evaluation in this setting particularly suited for our metrics. Methods that

cannot relate primitives across demonstrations (identify repeated structure) or reason about their temporal extents (identify segment structure) will be unable to recover the underlying procedure and should therefore be penalized. There may also be additional repeated structure *within* a time-series, for example if someone mixes cake batter, adds ingredients, and then mixes the cake batter again.

A closely related setting was tackled by the method outlined in Sener & Yao (2018), who assume that each time-series is a permutation of a shared bag of activities. However, as they acknowledge, a major limitation of their method is that they allow each activity to occur atmost once within a time-series, which limits repeated structure. Our problem setting allows for this possibility, which is often seen in real-world datasets *e.g.* the JIGSAWS surgical dataset (Gao et al., 2014) contains videos of surgeons doing multiple repetitions of primitives such as needle-passing or suturing. However, we do not allow the procedure to be permuted in our problem setting, which is a useful extension that we hope to explore in future work.

### 4.1 PRISM: PRocedure Identification with a Segmental Mixture model

We now describe a statistical model called PRISM for the procedure identification problem. PRISM is a generative hierarchical Bayesian model that explicitly separates reasoning about the procedure from its realization in each time-series in the dataset. It utilizes this to create a mixture distribution over primitives for every observation in the dataset. Fig 5 illustrates the model.

Unlike popular sequence models based on the Hidden Markov model, PRISM does not make a restrictive Markov assumption and can handle *any* procedure. And in contrast to a Gaussian mixture model, PRISM shares statistical strength across time-series by using a structured prior over the (discrete) local variables that generate each observation – amortizing the procedure across many time-series. As we show later in experiments, this structure can help PRISM outperform these methods, especially when the observations are noisy.

**Modeling the procedure.** PRISM assumes that each time-series $\mathbf{X}^i$ follows the same underlying procedure $\mathbf{p} = (p_1, \ldots, p_s)$, where $s$ is the number of steps in the procedure. $\mathbf{p}$ is a vector of discrete, global latent variables in the model. This implies that each time-series will split into $s$ segments, each segment corresponding to a single step in the procedure.

Next, let $|\Gamma_\mathcal{C}| = K$ be the number of cluster labels (primitives). Each step in the procedure will correspond to one of these $K$ labels *i.e.* $p_r \in [K], 1 \leq r \leq s$. Thus, $\mathbf{p}$ is an ordered sequence of high-level primitives *e.g.* to make tea, we would *boil water* $\rightarrow$ *get cup* $\rightarrow$ *pour water* and so on. Formally, we assume a generation process,

$$\boldsymbol{\lambda}_r \sim \mathrm{Dirichlet}(\alpha_1, \ldots, \alpha_K), \quad 1 \leq r \leq s \qquad p_r \sim \mathrm{Categorical}(\boldsymbol{\lambda}_r), \quad 1 \leq r \leq s$$

Each token in the procedure is independently generated from a separate Dirichlet prior $\boldsymbol{\lambda}_r$, which allows fine-grained control over the distribution of tokens at each step. Note that we do not disallow self-transitions, which can be utilized by the model to coalesce adjacent segments.

**Modeling realizations in each time-series.** Different time-series can be vastly different in how they implement the procedure *e.g.* they can be videos of different individuals making tea. These individuals, while following the same procedure, may spend varying amounts of time on each step (or even skip steps).

In PRISM, the realization of the procedure in the $i^{\mathrm{th}}$ time-series (having $m_i$ time-steps) is given by $\mathbf{w}_i = (w_{i,1}, \ldots, w_{i,m_i})$, a sorted sequence of discrete random variables. Each $w_{i,j} \in [s]$ is an index into the procedure $\mathbf{p}$. Its value dictates the step being followed in the procedure at the $j^{\mathrm{th}}$ time-step in the $i^{\mathrm{th}}$ time-series. We assume $\mathbf{w}_i$ is generated as follows,

$$\boldsymbol{\gamma} \sim \mathrm{Dirichlet}(\beta_1, \ldots, \beta_s) \qquad u_{i,j} \sim \mathrm{Categorical}(\boldsymbol{\gamma}), \quad 1 \leq i \leq n, 1 \leq j \leq m_i$$
$$\mathbf{w}_i = \mathrm{Sort}(u_{i,1}, \ldots, u_{i,m_i}), \quad 1 \leq i \leq n$$

Mathematically, to generate $\mathbf{w}_i$, we first generate an unordered bag of $m_i$ independent categorical variables $u_{i,j}$ and then sort them (in ascending order). Each $u_{i,j}$ is generated from the same Dirichlet prior $\boldsymbol{\gamma}$. Intuitively, if we use the stick-breaking metaphor for the Dirichlet distribution, $\boldsymbol{\gamma}$ is a prior over how long each token in the procedure is – $\gamma_r$ is the expected relative length of $\mathbf{p}_r$. Thus, $\boldsymbol{\gamma}$ allows us to impose a prior on the relative length of each step of the procedure, in every time-series. An equivalent view into our model is that each $\mathbf{w}_i$ is directly generated by a Multinomial distribution

with $m_i$ draws on probabilities dictated by $\boldsymbol{\gamma}$. However, we find that the view that we adopt yields a far more efficient inference scheme (see Appendix).

**Generating observations.** For every time-series, we can combine the common procedure $\mathbf{p}$ and the realization of the procedure $\mathbf{w}_i$ to recover a temporal clustering $\mathbf{z}_i$ for that time-series. We write this temporal clustering as $\mathbf{z}_i = (z_{i,1}, \ldots, z_{i,m_i})$ where $\mathbf{z}_{i,j} \in [K]$ is a discrete primitive.

For each primitive $k \in [K]$, we assume a Gaussian observation model $(\mu_k, \Sigma_k)$ (with a Normal-Inverse Wishart prior) shared by all time-series. PRISM then generates each $\mathbf{x}_j^i$ independently from its local assignment $z_{i,j}$ and the appropriate observation model. We use Gibbs sampling (see Appendix) to perform posterior inference of the latent variables $\left(\{p_r\}_{r\in[s]}, \{u_{i,j}\}_{i\in[n],j\in[m_i]}, \{\mu_k\}_{k\in[K]}, \{\Sigma_k\}_{k\in[K]}\right)$ in the model.

## 5 EXPERIMENTS WITH EVALUATION CRITERIA

We first compare our metrics to existing criteria by evaluating several competing methods on real datasets. We note that the methodology for validating our evaluation criteria is similar to that followed by Rosenberg & Hirschberg (2007), with the exception that we compare criteria on real-world data with a large number of methods. We compare to homogeneity (Hom), completeness (Com), NMI since they give the best combination of performance for clustering evaluation (Rosenberg & Hirschberg, 2007). We also compare to accuracy computed using the Munkres method since it is widely used *e.g.* (Sener & Yao, 2018; Zhou et al., 2008) and the adjusted Rand index (ARI) (Vinh et al., 2010).

**Methods.** We use several methods to generate temporal clusterings: Hierarchical Clustering (AGG); Hidden Markov Models (HMM); Hierarchical Dirichlet Process HMMs (HDP-HMM) (Teh et al., 2005); Sticky HDP-HMM (SHDP-HMM) (Fox et al., 2008a); HDP Hidden Semi-Markov Models (HDP-HSMM) (Johnson & Willsky, 2013); Switching Linear Dynamical Systems (SLDS) (Fox et al., 2008b); HDP-SLDS (Fox et al., 2008b); Sticky HDP-SLDS (SHDP-SLDS) (Fox et al., 2008b); Ordered Subspace Clustering (OSC) (Tierney et al., 2014); Temporal SC (TSC) (Li et al., 2015); Low-Rank SC (LRSC) (Vidal & Favaro, 2014); Sparse SC (SSC) (Elhamifar & Vidal, 2013) and Spatial SC (SPATSC) (Guo et al., 2013).

**Datasets.** We use 2 common benchmark datasets (Fox et al., 2008b; 2009; 2014; Zhou et al., 2008; 2013). BEES consists of 6 time-series, each recording the dancing behavior of a bee. Each time-series is built up of 3 primitive movements – *waggle*, *left-turn* and *right-turn* – repeated several times. For BEES, we treat each sequence as a separate dataset (BEES$_1$,...,BEES$_6$) and run each method on them individually, as done in Fox et al. (2008b). Here, we focus on results on BEES$_1$ but results on MOCAP6, a motion-capture dataset and all BEES datasets can be found in the Appendix.

**Results.** For each of our metrics, we first illustrate that they qualitatively capture the characteristics that they were designed for. Fig 6 visualizes the best and worst performing methods on BEES$_1$ for each of our metrics. In Fig 6a, SSC is worst on LASS-O since it heavily over-segments the time-series by introducing many transitions, while TSC introduces far fewer transitions. In Fig 6b, under-segmentation is penalized and we can see that AGG should be worse since it contains longer segments and far fewer transitions than SSC. Combining these ideas in Fig 6c yields that HDP-HSMM has the best LASS – observe that it contains fewer closely spaced transition points and creates a more even segmentation at the right level of granularity, explaining its score.

Fig 6e shows the result of evaluating with just the RSS – we can clearly see that SpatSC is able to capture far more repeated structure than SHDP-HMM, which is unable to label segments from the same ground-truth label consistently. Lastly, we see that SpatSC is judged to be the best method on BEES$_1$ by TSS in Fig 6f since it has the highest RSS and a SSS of $0.81$ which is close to the highest.

Improving scores on evaluation metrics is common practice to demonstrate performance gains due to a method. We now highlight cases where TSS disagrees with NMI, ARI and/or Munkres in the assessment of competing methods, since this discrepancy has significant impact on which method is picked. We find that such disagreements happen very often across *all* datasets. An illustrative example for BEES$_1$ is shown in Fig 6h: TSS and Munkres agree that SSC is better than HDP-HSMM, but NMI and ARI strongly disagree. Before diving into why this may be happening, we note that examining the RSS and SSS allows us to understand why TSS considers SSC to be better than HDP-HSMM – it captures more repeated structure. On the other hand Munkres, while it agrees with

| Dataset | Bees | | | | | | | Surgery | | | |
|---------|------|--------|--------|--------|--------|--------|---------|-------------|----------------|----------|----------|
| Method | Bees-1 (1) | Bees-2 (1) | Bees-3 (1) | Bees-4 (1) | Bees-5 (1) | Bees-6 (1) | Average | Knot-Tying (10) | Needle-Passing (10) | Suturing (10) | Suturing (5) |
| GMM | 61.21 | 62.38 | 44.74 | 63.86 | 74.79 | 66.44 | 62.24 | 25.12 | 21.93 | 34.75 | 49.03 |
| | 19.97 | 35.52 | 12.86 | 31.20 | 37.30 | 36.09 | 28.82 | 6.98 | 2.94 | 19.96 | 32.58 |
| HMM ($\alpha = 0.1$) | **66.43** | 70.80 | 52.35 | 78.20 | 87.38 | 73.62 | 71.46 | 22.94 | 25.44 | 38.91 | 49.92 |
| | **32.11** | 37.39 | 17.14 | 39.63 | 65.77 | 43.27 | 39.22 | 5.05 | 2.99 | 17.52 | 27.49 |
| HMM ($\alpha = 1.0$) | 65.00 | 69.16 | 53.28 | 73.75 | **87.45** | 80.54 | 71.53 | 24.48 | 22.72 | 38.97 | 48.72 |
| | 29.20 | 37.82 | 17.75 | 36.63 | **65.93** | 50.00 | 39.56 | 5.50 | 3.27 | 18.98 | 29.93 |
| HMM ($\alpha = 100.0$) | 58.69 | **78.28** | **53.56** | **81.51** | 85.85 | **86.45** | **74.06** | 25.02 | 22.91 | 37.73 | 47.32 |
| | 22.14 | **50.33** | **18.73** | **43.59** | 61.16 | **61.99** | **42.99** | 6.11 | 3.21 | 18.38 | 28.65 |
| Prism | 65.90 | 75.96 | **54.65** | 78.27 | 83.78 | 82.07 | 73.44 | **28.88** | **28.00** | **46.68** | **62.03** |
| | 25.04 | 46.01 | **18.26** | 41.57 | 56.11 | 53.72 | 40.12 | **9.75** | **15.47** | **33.26** | **51.54** |

Table 1: Comparison with baselines on BEES and the JIGSAWS surgical dataset. For a dataset, values in parentheses indicate the number of time-series. Upper entries are TSS and lower entries are NMI.

TSS, is unable to give us this insight. We find that NMI is highly sensitive to the number of clusters – SSC only uses 2 clusters, which means it is heavily penalized by NMI, in spite of capturing some repeated structure. ARI is less sensitive than NMI, but is far less interpretable.

Another example is shown in Fig 6g, where the TSS for both SLDS and HDP-HMM is comparable. Interestingly, NMI/ARI and Munkres are all highly skewed and disagree strongly on the better method. Since HDP-HMM has more clusters, the clusters are more homogenous and Hom is much higher, pushing up NMI. On the other hand, since Munkres finds a one-to-one correspondence between clusters and ground-truth labels, having more clusters means some cannot be matched. This causes the score to strongly downweigh suchh clusterings. Overall, our metrics provide both interpretability and the ability to analyze where the method is falling short.

Lastly, we conduct a sensitivity analysis for $\beta$ as follows: assume that the most appropriate value for evaluation is $\beta'$, but we instead use the default value of $\beta = 1.0$. Our main idea is to compute how well the ranking of methods under $\beta = 1.0$ captures that under $\beta'$. To do this, we use the Normalized Discounted Cumulative Gain (NDCG), a common criteria from information retrieval. Varying $\beta'$, we see in Fig 6i that $\beta = 1.0$ outputs similar rankings to a relatively wide range of other $\beta$ values, other than at the extremes. This confirms that most settings of $\beta$ will be captured well by our choice of $\beta$, unless there is a strong reason to prefer either the RSS or the SSS only.

## 6 EXPERIMENTS WITH PRISM

We now evaluate PRISM on complex datasets with noisy features, including video data. For PRISM, we set $\boldsymbol{\alpha} = 1.0, \boldsymbol{\beta} = 0.1$ (except BEES) and we set $K$ (number of mixture components) to equal the number of ground-truth labels for all methods. Results are averaged over 10 random seeds. Experimental details including hyperparameters can be found in the Appendix.

Before our main results, we briefly conduct a simple check on performance for the BEES dataset introduced earlier. Though this only has a single example per procedure, we find that our approach still does well (Table 1) compared to two simple baselines – all approaches use the same observation model with Gaussian emissions.

**Leveraging a fixed procedure.** We now evaluate the benefit of our approach on the JIGSAWS dataset (Gao et al., 2014), which consists of surgeons with varying skill demonstrating common procedures, such as suturing, needle-passing and knot-tying. We use kinematic features that correspond to the grippers controlled by the surgeon for all methods. For each surgical task, we select the 10 demonstrations that correspond to the 2 expert surgeons since they adhere most closely to the prescribed procedures. Even for a single expert, there is considerable variability across the 5 demonstrations they generate, so this is far from an idealized setting. Once again, all methods leverage the same observational model with the same features. Our baselines are two standard approaches – a Gaussian mixture model (GMM), and variants of a Bayesian Hidden Markov model.

As expected, PRISM outperforms these baselines by a considerable margin. On the suturing task, performance is improve by almost double-digits on both metrics, with performance gains also visible on the other tasks. Leveraging the common procedure allows PRISM to share statistical strength

|  | SENER (no background model) | PRISM (no background model) | SENER (background model) |
|---|---|---|---|
| *changing tire* | 25.0 | 31.2 | **33.9** |
| *coffee* | 22.1 | 26.9 | **29.0** |
| *cpr* | 18.1 | **28.1** | 24.9 |
| *jump car* | 5.8 | **22.4** | 15.0 |
| *repot plant* | 19.1 | **24.0** | **23.9** |

Table 2: Comparison with Sener & Yao (2018) on the INRIA dataset with the Munkres score.

across demonstrations and make consistent predictions where possible, whereas the other methods must rely more strongly on the observation model to infer the latent sequence. Without very cleanly separated features for different primitives, they can often be led astray.

We also experiment with a smaller subset of 5 demonstrations provided by a single expert on the suturing task. This subset of data has the least amount of variability in terms of the expert's procedure. Because of this, we see that PRISM gives an improvement of $\sim 18$ points on NMI and $\sim 10$ points on the TSS. To illustrate PRISM's sensitivity to hyperparameters, we vary $s$ and $K$ in Fig 7 on this single-expert suturing task. We find that PRISM's performance is quite stable with respect to both – this holds as long as $s$ is larger than the number of segments in ground-truth, and $K$ is close to the true number of ground-truth clusters.

**Noisy procedures with video data.** Next, we consider labeling demonstrations that are very noisy in their adherence to the procedure, with a great deal of variability in both the steps being followed, their order, as well as their relative lengths. Recent work by Sener & Yao (2018) achieved state-of-the-art performance without any form of supervision on two large video datasets – Breakfast actions (Kuehne et al., 2014) and INRIA instructional videos (Alayrac et al., 2016). We expect our method to perform worse on the Breakfast actions dataset, as it has no repeated structure, in which case our model has no advantages over this prior model, and indeed uses a simpler observation model. Perhaps encouragingly, in the Breakfast actions dataset, we find that our method only performs slightly worse than but close to Sener & Yao (2018), achieving a Munkres score of 33.5 compared to their score of 34.6. It is likely that this can be improved by tuning the number of segments for each procedure (we set $s = 20$ for all procedures).

However a key limitation of their work is that they do not model repeated structure. Therefore in domains with repeated structure we expect our approach to do well. Such structure is present in some of the activities in the INRIA dataset. We compare to Sener & Yao (2018)'s results on the INRIA dataset in Table 2. PRISM outperforms their best method on 3 out of the 5 activities despite not explicitly modeling background information, with a significant difference on *cpr* and *jump car*. On their comparable method which does not model background, PRISM is significantly better across all procedures. This is despite PRISM using a much weaker observation model (they simultaneously learn a feature representation for the videos) and no permutation model. An example for the *cpr* procedure is shown in Fig 1 and illustrates that unlike Sener & Yao (2018)'s method, PRISM can successfully recover repeated structure both within and across videos. We expect that incorporating a background model in PRISM could enable our approach to yield even stronger performance.

## 7 CONCLUSION

We developed PRISM, a hierarchical Bayesian model to identify latent procedures in time-series data. Results on several datasets show that PRISM can be used to learn procedural abstractions more effectively than competing methods. We also introduced new evaluation criteria for the problem of externally evaluating temporal clusterings. Our metrics address gaps in temporal clustering evaluation, and provide both interpretability and the potential for easier analysis. In future work, we hope to extend PRISM by incorporating richer observation models, and relaxing the assumption that a single strictly-ordered procedure is followed by all time-series. Links to code can be found in the Appendix.

## 8 ACKNOWLEDGMENTS

The authors would like to thank the Schmidt Foundation and a NSF Career Grant for their support.

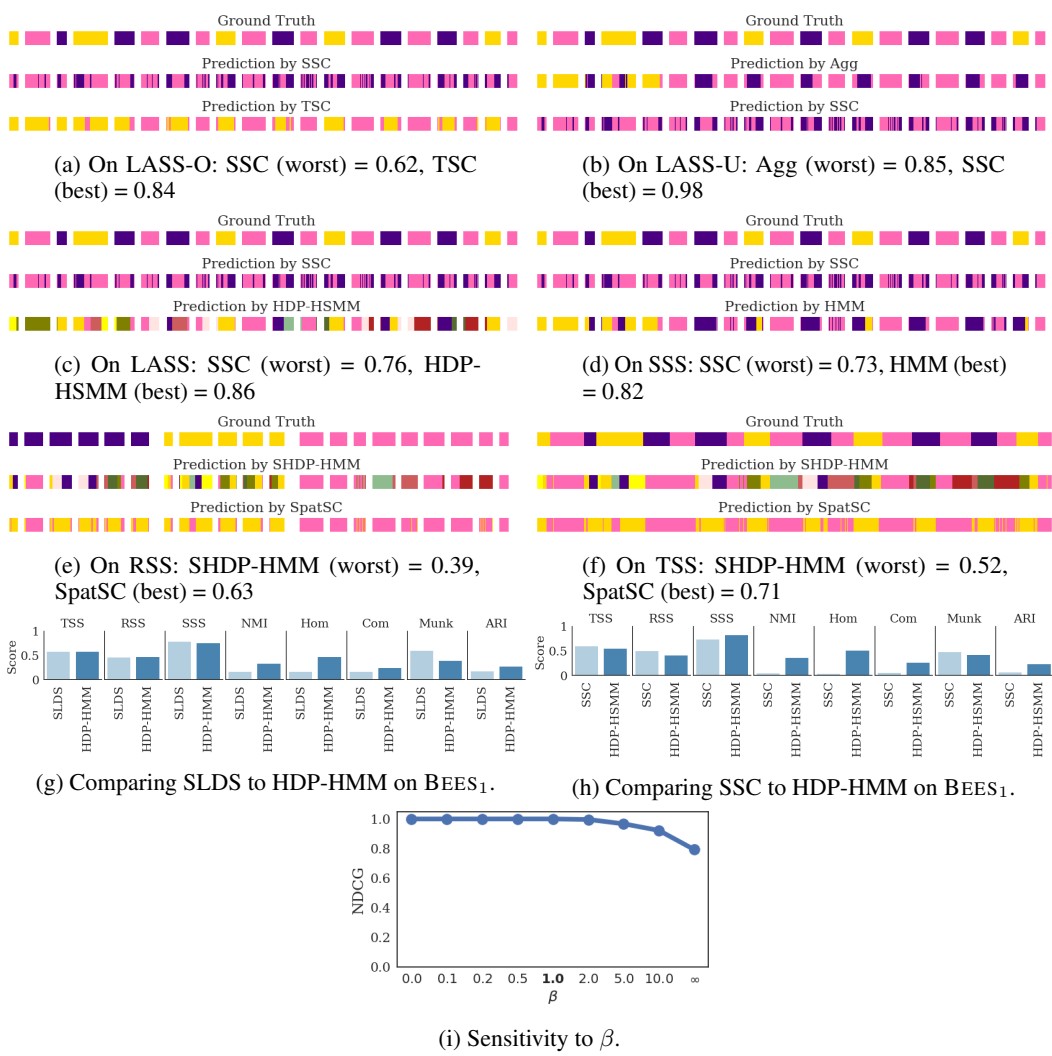

(a) On LASS-O: SSC (worst) = 0.62, TSC (best) = 0.84

(b) On LASS-U: Agg (worst) = 0.85, SSC (best) = 0.98

(c) On LASS: SSC (worst) = 0.76, HDP-HSMM (best) = 0.86

(d) On SSS: SSC (worst) = 0.73, HMM (best) = 0.82

(e) On RSS: SHDP-HMM (worst) = 0.39, SpatSC (best) = 0.63

(f) On TSS: SHDP-HMM (worst) = 0.52, SpatSC (best) = 0.71

(g) Comparing SLDS to HDP-HMM on BEES₁.

(h) Comparing SSC to HDP-HMM on BEES₁.

(i) Sensitivity to $\beta$.

Figure 6: (a)-(f) visualize the best and worst methods on our metrics; (g)-(h) show examples where we disagree with traditional criteria; (i) illustrates a sensitivity analysis with respect to the TSS tradeoff parameter $\beta$. All results are on the BEES₁ dataset.

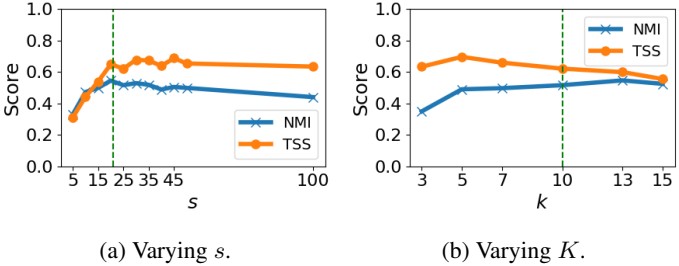

(a) Varying $s$.

(b) Varying $K$.

Figure 7: Effect of varying hyperparameters on model performance. Results are on the JIGSAWS suturing task. Green marks the value of the hyperparameter in ground-truth.

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

# A  ALGORITHM FOR REPEATED STRUCTURE SCORE

Detailed pseudocode for reproducing the computation of the RSS is given in Algorithm 1.

---
**Algorithm 1** Calculation of RSS
---
**Require:** Ground truth $\mathcal{G}$, temporal clustering $\mathcal{C}$, similarity function $\mathcal{H}$
 1: \\ Optional Pruning Step
 2: **for all** $\alpha \in \Gamma_{\mathcal{C}}$ **do** $\beta_{\alpha}^{*} \leftarrow \arg\max_{\beta \in \Gamma_{\mathcal{G}}} |\mathcal{G}[\beta] \cap \mathcal{C}[\alpha]|$ $\qquad\qquad$ ▷ find $\beta_{\alpha}^{*}$ which most overlaps $\alpha$
 3: **for all** $(a,b) \in \mathcal{S}(\mathcal{G},\beta)$, $\beta \in \Gamma_{\mathcal{G}}$ **do** $\qquad\qquad$ ▷ for every segment of every $\beta$ in $\mathcal{G}$
 4: $\quad$ $\mathcal{P}(\mathcal{C}_{a:b}), \mathcal{W}(\mathcal{C}_{a:b}) := (p_1, p_2, \ldots, p_r), (w_1, w_2, \ldots, w_r)$ $\qquad$ ▷ locate segment in $\mathcal{C}$
 5: $\quad$ $\tilde{w}_i \leftarrow w_i \times \mathbb{I}[\beta = \beta_{p_i}^{*}]$ $\qquad\qquad$ ▷ trim weights based on purity
 6: $\quad$ $\tilde{\mathcal{W}}(\mathcal{C}_{a:b}) \leftarrow (\tilde{w}_1, \tilde{w}_2, \ldots, \tilde{w}_r)$ $\qquad\qquad$ ▷ construct new weights
 7: \\ Repeated Structure Scoring
 8: score $\leftarrow \sum_{\beta \in \Gamma_{\mathcal{G}}} \sum_{(a_1,b_1),(a_2,b_2) \in \mathcal{S}(\mathcal{G},\beta)} \mathcal{H}(\mathcal{P}(\mathcal{C}_{a_1:b_1}), \mathcal{P}(\mathcal{C}_{a_2:b_2}), \tilde{\mathcal{W}}(\mathcal{C}_{a_1:b_1}), \tilde{\mathcal{W}}(\mathcal{C}_{a_2:b_2}))$
 9: $\qquad\qquad\qquad$ ▷ score every repeated structure pair for every ground-truth label $\beta$
10: max_score $\leftarrow 2 \sum_{\beta \in \Gamma_{\mathcal{G}}} |\mathcal{S}(\mathcal{G},\beta)| \sum_{(a,b) \in \mathcal{S}(\mathcal{G},\beta)} (b - a + 1)$
11: **return** RSS $\leftarrow \frac{\text{score}}{\text{max\_score}}$
---

# B  EXPERIMENTS WITH EVALUATION CRITERIA

The results presented for the evaluation criteria developed in this paper can be reproduced using code available at `https://github.com/StanfordAI4HI/ICLR2019_evaluating_discrete_temporal_structure`. The temporal clusterings predicted by each method, as well as ground-truth data is available at the link.

In addition, an implementation of only the newly proposed temporal clustering evaluation criteria can be found at `https://github.com/StanfordAI4HI/tclust-eval`.

For completeness, we include the result of evaluation across all methods, datasets and criteria.

| method metric | Agg | HDP-HMM | HDP-HSMM | HDP-SLDS | HMM | LRSC | OSC | SHDP-HMM | SHDP-SLDS | SLDS | SSC | SpatSC | TSC |
|---|---|---|---|---|---|---|---|---|---|---|---|---|---|
| ARI | 0.28 | 0.27 | 0.22 | 0.14 | 0.36 | 0.10 | 0.39 | 0.23 | 0.14 | 0.17 | 0.06 | 0.39 | 0.30 |
| Com | 0.25 | 0.24 | 0.25 | 0.13 | 0.30 | 0.13 | 0.44 | 0.25 | 0.13 | 0.16 | 0.05 | 0.44 | 0.34 |
| Hom | 0.24 | 0.46 | 0.51 | 0.13 | 0.30 | 0.13 | 0.28 | 0.50 | 0.13 | 0.16 | 0.03 | 0.28 | 0.22 |
| LASS | 0.84 | 0.78 | 0.86 | 0.82 | 0.86 | 0.83 | 0.84 | 0.84 | 0.82 | 0.84 | 0.76 | 0.84 | 0.86 |
| LASS-O | 0.82 | 0.65 | 0.80 | 0.74 | 0.81 | 0.73 | 0.78 | 0.77 | 0.80 | 0.81 | 0.62 | 0.78 | 0.84 |
| LASS-U | 0.85 | 0.97 | 0.92 | 0.91 | 0.92 | 0.95 | 0.92 | 0.94 | 0.86 | 0.88 | 0.98 | 0.92 | 0.87 |
| Munk | 0.61 | 0.39 | 0.41 | 0.58 | 0.68 | 0.50 | 0.64 | 0.39 | 0.58 | 0.59 | 0.47 | 0.64 | 0.60 |
| NMI | 0.24 | 0.33 | 0.36 | 0.13 | 0.30 | 0.13 | 0.35 | 0.35 | 0.13 | 0.16 | 0.04 | 0.35 | 0.27 |
| RSS | 0.55 | 0.46 | 0.40 | 0.53 | 0.57 | 0.45 | 0.63 | 0.39 | 0.49 | 0.45 | 0.49 | 0.63 | 0.62 |
| SSS | 0.78 | 0.75 | 0.81 | 0.76 | 0.82 | 0.79 | 0.81 | 0.81 | 0.76 | 0.78 | 0.73 | 0.81 | 0.81 |
| TSS | 0.65 | 0.57 | 0.54 | 0.62 | 0.67 | 0.57 | 0.71 | 0.52 | 0.59 | 0.58 | 0.59 | 0.71 | 0.70 |

Table 3: Results on $\text{BEES}_1$ across all evaluation criteria and methods.

| method metric | Agg | HDP-HMM | HDP-HSMM | HDP-SLDS | HMM | LRSC | OSC | SHDP-HMM | SHDP-SLDS | SLDS | SSC | SpatSC | TSC |
|---|---|---|---|---|---|---|---|---|---|---|---|---|---|
| ARI | 0.26 | 0.32 | 0.28 | 0.14 | 0.61 | 0.19 | 0.44 | 0.25 | 0.12 | 0.10 | 0.05 | 0.44 | 0.30 |
| Com | 0.27 | 0.33 | 0.28 | 0.16 | 0.54 | 0.19 | 0.51 | 0.29 | 0.13 | 0.11 | 0.04 | 0.51 | 0.32 |
| Hom | 0.28 | 0.66 | 0.53 | 0.16 | 0.54 | 0.20 | 0.33 | 0.58 | 0.20 | 0.10 | 0.02 | 0.33 | 0.41 |
| LASS | 0.79 | 0.77 | 0.87 | 0.83 | 0.89 | 0.79 | 0.84 | 0.84 | 0.83 | 0.82 | 0.76 | 0.84 | 0.87 |
| LASS-O | 0.69 | 0.64 | 0.82 | 0.80 | 0.91 | 0.67 | 0.78 | 0.77 | 0.77 | 0.82 | 0.62 | 0.78 | 0.84 |
| LASS-U | 0.94 | 0.96 | 0.92 | 0.87 | 0.88 | 0.96 | 0.91 | 0.93 | 0.90 | 0.82 | 0.98 | 0.91 | 0.90 |
| Munk | 0.64 | 0.38 | 0.44 | 0.56 | 0.85 | 0.60 | 0.68 | 0.40 | 0.37 | 0.49 | 0.49 | 0.68 | 0.54 |
| NMI | 0.27 | 0.47 | 0.38 | 0.16 | 0.54 | 0.20 | 0.41 | 0.41 | 0.16 | 0.10 | 0.03 | 0.41 | 0.37 |
| RSS | 0.51 | 0.60 | 0.42 | 0.44 | 0.77 | 0.40 | 0.73 | 0.57 | 0.40 | 0.41 | 0.49 | 0.73 | 0.50 |
| SSS | 0.75 | 0.75 | 0.82 | 0.76 | 0.86 | 0.76 | 0.80 | 0.79 | 0.76 | 0.74 | 0.74 | 0.80 | 0.82 |
| TSS | 0.61 | 0.67 | 0.56 | 0.56 | 0.81 | 0.52 | 0.76 | 0.66 | 0.53 | 0.53 | 0.59 | 0.76 | 0.62 |

Table 4: Results on $\text{BEES}_2$ across all evaluation criteria and methods.

| method / metric | Agg | HDP-HMM | HDP-HSMM | HDP-SLDS | HMM | LRSC | OSC | SHDP-HMM | SLDS | SSC | SpatSC | TSC |
|---|---|---|---|---|---|---|---|---|---|---|---|---|
| ARI | 0.00 | 0.12 | 0.14 | 0.17 | 0.16 | 0.33 | 0.08 | 0.15 | 0.33 | 0.05 | 0.16 | 0.04 |
| Com | 0.02 | 0.23 | 0.23 | 0.14 | 0.18 | 0.32 | 0.14 | 0.24 | 0.29 | 0.13 | 0.20 | 0.08 |
| Hom | 0.02 | 0.47 | 0.49 | 0.19 | 0.19 | 0.21 | 0.15 | 0.53 | 0.30 | 0.08 | 0.34 | 0.10 |
| LASS | 0.80 | 0.79 | 0.82 | 0.81 | 0.79 | 0.80 | 0.79 | 0.80 | 0.77 | 0.78 | 0.80 | 0.78 |
| LASS-O | 0.74 | 0.69 | 0.75 | 0.75 | 0.82 | 0.77 | 0.68 | 0.71 | 0.76 | 0.66 | 0.71 | 0.76 |
| LASS-U | 0.86 | 0.94 | 0.90 | 0.88 | 0.78 | 0.83 | 0.94 | 0.91 | 0.78 | 0.94 | 0.93 | 0.80 |
| Munk | 0.41 | 0.33 | 0.33 | 0.48 | 0.46 | 0.68 | 0.48 | 0.32 | 0.64 | 0.47 | 0.38 | 0.36 |
| NMI | 0.02 | 0.33 | 0.34 | 0.16 | 0.18 | 0.26 | 0.15 | 0.36 | 0.29 | 0.10 | 0.26 | 0.09 |
| RSS | 0.28 | 0.42 | 0.30 | 0.56 | 0.45 | 0.65 | 0.41 | 0.37 | 0.61 | 0.44 | 0.40 | 0.28 |
| SSS | 0.73 | 0.74 | 0.77 | 0.74 | 0.74 | 0.75 | 0.73 | 0.75 | 0.71 | 0.74 | 0.75 | 0.72 |
| TSS | 0.41 | 0.54 | 0.43 | 0.64 | 0.56 | 0.70 | 0.53 | 0.50 | 0.66 | 0.55 | 0.52 | 0.40 |

Table 5: Results on BEES$_3$ across all evaluation criteria and methods.

| method / metric | Agg | HDP-HMM | HDP-HSMM | HDP-SLDS | HMM | LRSC | OSC | SHDP-HMM | SHDP-SLDS | SLDS | SSC | SpatSC | TSC |
|---|---|---|---|---|---|---|---|---|---|---|---|---|---|
| ARI | 0.28 | 0.30 | 0.52 | 0.25 | 0.54 | 0.12 | 0.18 | 0.31 | 0.29 | 0.28 | 0.02 | 0.18 | 0.17 |
| Com | 0.30 | 0.28 | 0.42 | 0.16 | 0.48 | 0.13 | 0.17 | 0.26 | 0.27 | 0.27 | 0.02 | 0.17 | 0.16 |
| Hom | 0.31 | 0.53 | 0.62 | 0.16 | 0.45 | 0.14 | 0.11 | 0.50 | 0.18 | 0.17 | 0.01 | 0.11 | 0.17 |
| LASS | 0.87 | 0.84 | 0.87 | 0.81 | 0.87 | 0.81 | 0.81 | 0.83 | 0.84 | 0.84 | 0.77 | 0.81 | 0.78 |
| LASS-O | 0.86 | 0.79 | 0.84 | 0.71 | 0.85 | 0.73 | 0.78 | 0.76 | 0.77 | 0.77 | 0.67 | 0.78 | 0.82 |
| LASS-U | 0.88 | 0.91 | 0.91 | 0.94 | 0.90 | 0.91 | 0.85 | 0.91 | 0.92 | 0.92 | 0.92 | 0.85 | 0.75 |
| Munk | 0.52 | 0.40 | 0.63 | 0.56 | 0.81 | 0.52 | 0.57 | 0.40 | 0.62 | 0.61 | 0.43 | 0.57 | 0.50 |
| NMI | 0.30 | 0.38 | 0.51 | 0.16 | 0.46 | 0.13 | 0.14 | 0.36 | 0.22 | 0.22 | 0.02 | 0.14 | 0.17 |
| RSS | 0.51 | 0.43 | 0.72 | 0.69 | 0.85 | 0.40 | 0.61 | 0.50 | 0.71 | 0.70 | 0.49 | 0.61 | 0.51 |
| SSS | 0.82 | 0.79 | 0.83 | 0.74 | 0.81 | 0.77 | 0.75 | 0.77 | 0.79 | 0.78 | 0.73 | 0.75 | 0.72 |
| TSS | 0.62 | 0.56 | 0.77 | 0.71 | 0.83 | 0.52 | 0.68 | 0.60 | 0.75 | 0.74 | 0.58 | 0.68 | 0.60 |

Table 6: Results on BEES$_4$ across all evaluation criteria and methods.

| method / metric | Agg | HDP-HMM | HDP-HSMM | HDP-SLDS | HMM | LRSC | OSC | SHDP-HMM | SHDP-SLDS | SLDS | SSC | SpatSC | TSC |
|---|---|---|---|---|---|---|---|---|---|---|---|---|---|
| ARI | 0.14 | 0.37 | 0.49 | 0.24 | 0.68 | 0.31 | 0.31 | 0.24 | 0.24 | 0.15 | 0.10 | 0.31 | 0.15 |
| Com | 0.21 | 0.38 | 0.50 | 0.30 | 0.66 | 0.41 | 0.31 | 0.30 | 0.24 | 0.14 | 0.15 | 0.31 | 0.17 |
| Hom | 0.19 | 0.64 | 0.67 | 0.19 | 0.66 | 0.26 | 0.30 | 0.57 | 0.18 | 0.14 | 0.09 | 0.30 | 0.11 |
| LASS | 0.82 | 0.84 | 0.88 | 0.85 | 0.90 | 0.85 | 0.84 | 0.82 | 0.86 | 0.83 | 0.83 | 0.84 | 0.84 |
| LASS-O | 0.79 | 0.76 | 0.86 | 0.79 | 0.90 | 0.88 | 0.80 | 0.73 | 0.81 | 0.74 | 0.75 | 0.80 | 0.83 |
| LASS-U | 0.86 | 0.93 | 0.91 | 0.93 | 0.91 | 0.83 | 0.89 | 0.93 | 0.92 | 0.93 | 0.92 | 0.89 | 0.84 |
| Munk | 0.57 | 0.51 | 0.67 | 0.56 | 0.88 | 0.61 | 0.68 | 0.33 | 0.54 | 0.53 | 0.48 | 0.68 | 0.50 |
| NMI | 0.20 | 0.49 | 0.58 | 0.24 | 0.66 | 0.32 | 0.30 | 0.41 | 0.21 | 0.14 | 0.11 | 0.30 | 0.14 |
| RSS | 0.58 | 0.78 | 0.77 | 0.61 | 0.90 | 0.47 | 0.73 | 0.63 | 0.59 | 0.56 | 0.57 | 0.73 | 0.58 |
| SSS | 0.74 | 0.77 | 0.83 | 0.81 | 0.86 | 0.80 | 0.77 | 0.75 | 0.81 | 0.76 | 0.78 | 0.77 | 0.75 |
| TSS | 0.65 | 0.78 | 0.80 | 0.69 | 0.88 | 0.59 | 0.75 | 0.68 | 0.68 | 0.64 | 0.65 | 0.75 | 0.66 |

Table 7: Results on BEES$_5$ across all evaluation criteria and methods.

| method / metric | Agg | HDP-HMM | HDP-HSMM | HDP-SLDS | HMM | LRSC | OSC | SHDP-HMM | SHDP-SLDS | SLDS | SSC | SpatSC | TSC |
|---|---|---|---|---|---|---|---|---|---|---|---|---|---|
| ARI | 0.21 | 0.50 | 0.50 | 0.26 | 0.68 | 0.13 | 0.38 | 0.25 | 0.24 | 0.27 | 0.07 | 0.38 | 0.25 |
| Com | 0.22 | 0.42 | 0.43 | 0.18 | 0.62 | 0.15 | 0.43 | 0.32 | 0.26 | 0.23 | 0.07 | 0.43 | 0.32 |
| Hom | 0.20 | 0.70 | 0.70 | 0.20 | 0.62 | 0.27 | 0.28 | 0.65 | 0.17 | 0.20 | 0.04 | 0.28 | 0.51 |
| LASS | 0.82 | 0.85 | 0.87 | 0.79 | 0.89 | 0.74 | 0.85 | 0.82 | 0.81 | 0.81 | 0.77 | 0.85 | 0.80 |
| LASS-O | 0.81 | 0.80 | 0.84 | 0.70 | 0.88 | 0.60 | 0.85 | 0.74 | 0.75 | 0.73 | 0.67 | 0.85 | 0.74 |
| LASS-U | 0.82 | 0.90 | 0.90 | 0.91 | 0.90 | 0.95 | 0.86 | 0.92 | 0.89 | 0.91 | 0.90 | 0.86 | 0.87 |
| Munk | 0.52 | 0.64 | 0.63 | 0.54 | 0.89 | 0.33 | 0.63 | 0.36 | 0.57 | 0.64 | 0.48 | 0.63 | 0.44 |
| NMI | 0.21 | 0.54 | 0.55 | 0.19 | 0.62 | 0.20 | 0.35 | 0.46 | 0.21 | 0.21 | 0.05 | 0.35 | 0.40 |
| RSS | 0.49 | 0.68 | 0.66 | 0.66 | 0.89 | 0.39 | 0.73 | 0.53 | 0.63 | 0.69 | 0.49 | 0.73 | 0.57 |
| SSS | 0.75 | 0.80 | 0.83 | 0.72 | 0.84 | 0.69 | 0.79 | 0.76 | 0.76 | 0.76 | 0.71 | 0.79 | 0.73 |
| TSS | 0.59 | 0.73 | 0.73 | 0.69 | 0.87 | 0.50 | 0.76 | 0.62 | 0.69 | 0.72 | 0.58 | 0.76 | 0.64 |

Table 8: Results on BEES$_6$ across all evaluation criteria and methods.

# C  DETAILS FOR PRISM

This section contains details related to both the modeling and inference performed by PRISM.

| method
metric | Agg | HDP-HMM | HDP-HSMM | HDP-SLDS | HMM | LRSC | OSC | SHDP-HMM | SHDP-SLDS | SLDS | SSC | SpatSC | TSC |
|---|---|---|---|---|---|---|---|---|---|---|---|---|---|
| ARI | 0.26 | 0.47 | 0.35 | 0.29 | 0.46 | 0.24 | 0.17 | 0.49 | 0.33 | 0.48 | 0.02 | 0.17 | 0.16 |
| Com | 0.45 | 0.62 | 0.54 | 0.56 | 0.59 | 0.51 | 0.41 | 0.63 | 0.56 | 0.60 | 0.35 | 0.41 | 0.45 |
| Hom | 0.48 | 0.64 | 0.53 | 0.32 | 0.61 | 0.28 | 0.28 | 0.64 | 0.41 | 0.56 | 0.13 | 0.28 | 0.21 |
| LASS | 0.77 | 0.87 | 0.91 | 0.85 | 0.85 | 0.80 | 0.79 | 0.87 | 0.84 | 0.84 | 0.81 | 0.79 | 0.81 |
| LASS-O | 0.64 | 0.78 | 0.90 | 0.75 | 0.76 | 0.69 | 0.66 | 0.79 | 0.74 | 0.74 | 0.75 | 0.66 | 0.77 |
| LASS-U | 0.98 | 0.97 | 0.92 | 0.97 | 0.97 | 0.95 | 0.98 | 0.96 | 0.97 | 0.98 | 0.89 | 0.98 | 0.86 |
| Munk | 0.40 | 0.62 | 0.47 | 0.46 | 0.59 | 0.47 | 0.36 | 0.59 | 0.47 | 0.61 | 0.32 | 0.36 | 0.30 |
| NMI | 0.47 | 0.63 | 0.53 | 0.42 | 0.60 | 0.38 | 0.34 | 0.63 | 0.48 | 0.58 | 0.21 | 0.34 | 0.31 |
| RSS | 0.38 | 0.57 | 0.47 | 0.55 | 0.56 | 0.42 | 0.39 | 0.63 | 0.50 | 0.62 | 0.27 | 0.39 | 0.35 |
| SSS | 0.77 | 0.86 | 0.89 | 0.84 | 0.84 | 0.80 | 0.79 | 0.86 | 0.83 | 0.83 | 0.80 | 0.79 | 0.79 |
| TSS | 0.51 | 0.69 | 0.61 | 0.67 | 0.68 | 0.55 | 0.52 | 0.73 | 0.62 | 0.71 | 0.41 | 0.52 | 0.49 |

Table 9: Results on MOCAP6 across all evaluation criteria and methods.

## C.1 GENERATION IN PRISM

The generative process used in the hierarchical Bayesian model is given below,

$$\boldsymbol{\lambda}_r \sim \mathrm{Dir}(\alpha_1, \dots, \alpha_k), \quad 1 \le r \le s$$
$$\boldsymbol{\gamma} \sim \mathrm{Dir}(\beta_1, \dots, \beta_s)$$
$$p_r \sim \mathrm{Cat}(\boldsymbol{\lambda}_r), \quad 1 \le r \le s$$
$$u_{i,j} \sim \mathrm{Cat}(\boldsymbol{\gamma}), \quad 1 \le i \le n, 1 \le j \le m_i$$
$$\mathbf{w}_i = \mathrm{Sort}(u_{i,1}, \dots, u_{i,m_i}), \quad 1 \le i \le n$$
$$\mathbf{z}_i = \mathrm{TemporalClustering}(\mathbf{p}, \mathbf{w}_i), \quad 1 \le i \le n$$
$$(\mu_k, \Sigma_k) \sim \mathrm{NIW}(\mu_0, \Sigma_0, \kappa_0, \nu_0), \quad 1 \le k \le K$$
$$\mathbf{x}_{i,j} \sim \mathrm{Normal}(\mu_{z_{i,j}}, \Sigma_{z_{i,j}}), \quad 1 \le i \le n, 1 \le j \le m_i$$

## C.2 INFERENCE IN PRISM

We use Gibbs sampling to perform posterior inference of the latent variables $(\{p_r\}_{r \in [s]}, \{u_{i,j}\}_{i \in [n], j \in [m_i]}, \{\mu_k\}_{k \in [K]}, \{\Sigma_k\}_{k \in [K]})$ in the model.

**Sampling** $u_{i,j}$**.** Let $\Phi = \left(\{\mu_k\}_{k \in [K]}, \{\Sigma_k\}_{k \in [K]}\right)$, and write the complete conditional for $u_{i,j}$,

$$\mathrm{Pr}(u_{i,j} = r | \mathbf{u}_{-i}, \mathbf{u}_i^{-j}, \dots) \propto \mathrm{Pr}(u_{i,j} = r | \boldsymbol{\beta}, \mathbf{u}_{-i}, \mathbf{u}_i^{-j}) \prod_{j=1}^{m_i} \mathrm{Pr}(\mathbf{x}_{i,j} | u_{i,j} = r, \mathbf{u}_i^{-j}, \mathbf{p}, \Phi)$$

where in the first term we have collapsed the Dirichlet prior $\boldsymbol{\gamma}$. The main difficulty is in the data-likelihood terms $\mathrm{Pr}(\mathbf{x}_{i,j} | u_{i,j} = r, \dots)$, where we must reason about the effect of changing $u_{i,j}$. To reason about this effect, we must look at $\mathbf{w}_i^{(r)} = \mathrm{Sort}(u_{i,j} = r, \mathbf{u}_i^{-j})$, combine it with $\mathbf{p}$ to yield $\mathbf{z}_i^{(r)}$ and then compute the observation-likelihood terms. Due to the sorting step that yields $\mathbf{w}_i^{(r)}$, changing a single $u_{i,j}$ affects the entire sequence $\mathbf{z}_i^{(r)}$. Naively, we could compute $\mathrm{Pr}(\mathbf{x}_{i,j} | u_{i,j} = r, \dots)$ for every $j \in [m_i], r \in [s]$ but this would be extremely-inefficient (compute $s \cdot m_i$ terms).

Instead, we note that the assignments $\mathbf{z}_i^{(r)}$ will be almost identical for every $r \in [s]$ since $\mathbf{u}_i^{-j}$ is held fixed and only $u_{i,j}$ is varied. The only points at which the $\mathbf{z}_i^{(r)}$ are not identical are those right before and right after segment boundaries – $O(s)$ such locations. We only need to compute the data-likelihood at these locations since no other locations are changed by sampling $u_{i,j}$. Using this insight, we only need to compute $s^2$ terms, a big reduction since we expect $s << m_i$. We sample all the $u_{i,j}$'s sequentially using this update.

**Sampling** $p_r$**.** To sample $p_r$, we write the complete conditional,

$$\mathrm{Pr}(p_r = k | \mathbf{p}_{-r}, \boldsymbol{\alpha}, \dots) \propto \mathrm{Pr}(p_r = k | \boldsymbol{\alpha}) \prod_{i=1}^{n} \prod_{j=1}^{m_i} \mathrm{Pr}(\mathbf{x}_{i,j} | p_r = k, \mathbf{p}_{-r}, \mathbf{w}_i, \Phi)$$

Let $\mathbf{z}_i^{(k)}$ be the temporal clustering generated by combining $(p_r = k, \mathbf{p}_{-r})$ with $\mathbf{w}_i$. Note that in each time-series, setting $p_r = k$ affects only those observations that are assigned to $p_r$ by $\mathbf{w}_i$. We

thus need to compute the observation-likelihood of only these terms,

$$\Pr(p_r = k | \mathbf{p}_{-r}, \boldsymbol{\alpha}, \dots) \propto \Pr(p_r = k | \boldsymbol{\alpha}) \prod_{i=1}^{n} \prod_{\substack{j=1 \\ z_{i,j}=k}}^{m_i} \Pr(\mathbf{x}_{i,j} | \mu_k, \Sigma_k)$$

We sample the $p_r$'s sequentially using this update.

**Sampling** $\mu_k, \Sigma_k$. Sampling for the observation model can be done directly from the posterior of the Normal-Wishart Inverse distribution. We set uninformed priors for the observation model, $\mu_0 = \mathbf{0}_d, \Sigma_0 = \mathbf{I}_{d \times d}, \kappa_0 = 1, \nu_0 = d + 2$.

## D    Experiments with Prism

The results presented for PRISM can be reproduced using code available at `https://github.com/StanfordAI4HI/ICLR2019_prism`, which contains a Python implementation of PRISM.

### D.1    Datasets

We now provide details for preprocessing each dataset, with data loaders available at the linked code.

**JIGSAWS.** For the JIGSAWS dataset, we use the first 38 kinematic features. We subsample every 3rd timestep. To preprocess the data, we perform Principal Components Analysis (PCA) for dimensionality reduction to 12 features. We then stack features from 10 time-steps and perform PCA again to reduce dimensionality to 10 features. Lastly, we standardize features to have zero mean and unit variance.

**Breakfast actions.** We use the Fisher vector features taken from the 'stereo01' camera without any other preprocessing. We use the coarse segmentation for evaluation.

**INRIA instructional videos.** We use PCA to reduce the dimensionality of the features to $64$.

**Bees.** We use 4 features $- x, y, \cos\theta$ and $\sin\theta$.

### D.2    Experiment details

We use the hyperparameter settings given in Table 10 for PRISM. The Bayesian HMM has a single hyperparameter $\alpha$ which represents the hyperparameter for the Dirichlet prior over the transition matrix.

| Hyperparameter | Surgical | Breakfast | INRIA | Bees |
|:---:|:---:|:---:|:---:|:---:|
| $\alpha$ | 1.0 | 1.0 | 1.0 | 1.0 |
| $\beta$ | 0.1 | 0.1 | 0.1 | 1.0 |
| $s$ | 25 | 20 | 10 | 200 |
| $k$ | ground-truth | ground-truth | ground-truth | ground-truth |

Table 10: Hyperparameter settings used for PRISM experiments.

### D.3    Results on Breakfast Actions

Detailed results for the Breakfast actions dataset are summarized in Table 11.

| Procedure | Cereals | Coffee | Fried Egg | Juice | Milk | Pancakes | Salad | Sandwiches | Scrambled Eggs | Tea |
|:---:|:---:|:---:|:---:|:---:|:---:|:---:|:---:|:---:|:---:|:---:|
| NMI | 0.10 | 0.14 | 0.17 | 0.31 | 0.16 | 0.35 | 0.11 | 0.21 | 0.25 | 0.10 |
| Munkres | 0.36 | 0.30 | 0.29 | 0.43 | 0.40 | 0.34 | 0.29 | 0.35 | 0.29 | 0.29 |
| TSS | 0.43 | 0.45 | 0.51 | 0.57 | 0.47 | 0.62 | 0.38 | 0.49 | 0.47 | 0.40 |

Table 11: Procedure-wise breakdown of results on the Breakfast actions dataset.

| Method | NMI | TSS |
|---|---|---|
| GMM | 75.21 | 73.37 |
| HMM ($\alpha = 0.1$) | 47.68 | 72.50 |
| HMM ($\alpha = 1.0$) | 56.45 | 72.39 |
| HMM ($\alpha = 100.0$) | 59.32 | 51.31 |
| Prism ($\beta = 0.1$) | **79.04** | **82.77** |

Table 12: Comparison of methods in their ability to extract non-Markov procedural structure on 2-dimensional simulated data.

### D.4 NON-MARKOV SIMULATION STUDY

We generate 10 time-series using the following simple non-Markov sequence,

$$AABBAACCAADDAAEEAAFFAAGGAAHHAAAAAAAA$$

There are 8 distinct latent tokens in the sequence, and we generate noisy Gaussian 2-dimensional observations for each such token, using a mean of $(i, i)$ for the $i^{\text{th}}$ token, and a standard-deviation of 0.35 in either dimension.

We compare the Hidden Markov model variants to PRISM and the Gaussian mixture model. All methods use the same observational model with Gaussian emissions.

We expect that the main advantage of using an HMM vis-a-vis a GMM for recovering the latent temporal clustering, is in cases where the observations are noisy. In such cases, the learned transition dynamics can override the (noisy) observational likelihood to yield the correct sequence of tokens, as long as the procedure is Markov. However, this benefit should disappear in cases where the procedure is not Markov, like the sequence constructed by us above.

In contrast to these methods, PRISM makes no Markov assumption about the underlying procedure. This allows it to learn any procedure, Markov or otherwise. In addition, by sharing this procedure across multiple time-series, PRISM can avoid being led astray by noisy observations (unlike the GMM, which relies purely on the observation to infer the local latent token).

This argument is confirmed by the results in Table 12. PRISM outperforms all methods on both evaluation metrics, while the HMM variants suffer due to their inability to abandon the Markov assumption that is baked into the model.

