# OpenReview forum: "Learning Procedural Abstractions and Evaluating Discrete Latent Temporal Structure"
_ICLR.cc/2019/Conference_

### Official Review · AnonReviewer1 · 2018-10-29
**An interesting contribution on temporal clustering which consists off a new quality criterion and off a new model.**

**Rating:** 7
**Confidence:** 3

**Review:**

This paper describes two distinct contributions: a new compound criterion for comparing a temporal clustering to a ground truth clustering and a new bayesian temporal clustering method. Globally the paper is clear and well illustrated.
1) About the new criterion:
*pros: *
 a) as clearly pointed out by the authors, using standard non temporal clustering comparison metrics for temporal clustering evaluation is in a way "broken by design" as standard metrics disregard the very specificity of the problem. Thus the introduction of metrics that take explicitly into account time is extremely important.
 b) the proposed criterion combines two parts that are very important: finding the length of the stable intervals (i.e. intervals whose instants are all classified into a single cluster) and finding the sequence of labels.
*cons:*
 a) while the criterion seems new it is also related to criteria used in the segmentation literature (see among many other https://doi.org/10.1080/01621459.2012.737745) and it would have been a good idea to discuss the relation between temporal clustering and segmentation, even briefly.
b) the reliance on a tradeoff parameter in the final criterion is a major problem: how shall one chose the parameter (more on this below)? The paper does not explore the effect of modifying the parameter.
c) in the experimental section, TSS is mostly compared to NMI and to optimal matching (called Munkres here). Even considering the full list of criteria in the appendix, the normalized rand index (NRI) seems to be missing. This is a major oversight as the NRI is very adapted to comparing clusterings with different number of clusters, contrarily to NMI. In addition, the authors claim that optimal matching is completely opaque and difficult to analyse, while on the contrary it gives a proper way of comparing clusters from different clusterings, enabling fine grain analysis.

2) about the new model
*pros*:
 a) as far as I know, this is indeed a new model
 b) the way the model is structured emphasizes segmentation rather than temporal dependency: the so called procedure is arbitrary and no dependency is assumed from one segment to another. In descriptive analysis this is highly desirable (as opposed to say HMM which focuses on temporal dependencies).
*cons*
a) the way the length of the segments in the sequence are generated (with sorting) this a bit convolved. Why not generating directly those lengths? What is the distribution of those lengths under the sampling model? Is this adapted?
b) I find the experimental evaluation acceptable but a bit poor. In particular, nothing is said on how a practitioner would tune the parameters. I can accept that the model will be rather insensitive to hyper-parameters alpha and beta, but I've serious doubt about the number of clusters, especially as the evaluation is done here in the best possible setting. In addition, the other beta parameter (of TSS) is not studied.

Minor point:
- do not use beta for two different things (the balance in TSS and the prior parameter in the model)

---

> ### Author Response · Authors · 2018-11-27
> **Author Response**
>
> Thank you for your feedback! Here’s a response to the concerns that you raised,
>
>
> Existing segmentation criteria:
> We have included additional descriptive details in the revision in Section 2 & 3 on the relationship between temporal clustering and segmentation. Thank you for pointing us to the paper by Killick, Fearnhead, & Eckley (2011). We have included a discussion of this in the paper in Section 3. Their work uses criteria that are not very similar to those that we have proposed -- e.g. a criteria from their work (which is common in changepoint detection) is to evaluate whether a changepoint occurred close to (within some tolerance interval) one in ground-truth and measure the precision/recall. However, this is (i) sensitive to the tolerance interval, which is problem specific; (ii) an all or nothing metric which cannot distinguish small degradations or changes in the temporal clustering, unlike our approach.
>
>
> Reliance on a tradeoff parameter:
> Currently, the paper studies 3 settings of this \beta parameter -- 0, 1 and \infty. Our hope with these settings was to expose the behavior of the constituent metrics in a problem-agnostic way.
>
> Our original goal was to draw on the widely used criteria that have been designed for evaluating clusterings (Rosenberg and Hirschberg 2007) and consider them in the temporal setting. In doing so we encountered a multi-objective problem in how to weigh the new criteria that we designed (RSS and SSS). Our solution to introduce a tradeoff parameter (\beta) follows the approach laid out in the key paper on (non-temporal) clustering evaluation by Rosenberg and Hirschberg (2007) that introduces the V-measure for clustering evaluation. That paper includes a tradeoff parameter between completeness and homogeneity (the constituent criteria for V-measure) with the harmonic mean (\beta=1) kept as the default, allowing them to “prioritize one criterion over another, depending on the clustering task and goals.”
>
> Based on your and other reviewers’ excellent feedback to further explore how these settings impact the resulting metric, we have also introduced a new sensitivity analysis for the tradeoff parameter. While determining the right value of \beta for evaluation is a function of the problem and end-goal (e.g. repeated structure may have no importance in changepoint segmentation, and should be disregarded), we show that \beta=1 is a problem-agnostic compromise that works well in practice. To this end, our sensitivity analysis answers the following question: suppose \beta’ =/= 1.0 is the right value of the tradeoff parameter for a particular problem -- how similar does the metric perform (in terms of the ranking of methods) by using \beta=1.0? We explored this issue and show the results in Figure 6i in our revised paper. As expected,  at extreme values of \beta (0 or \infty) the methods may be ranked differently than by \beta=1.0. However, encouragingly, a large range of \beta values can be approximated well by using \beta=1.0 . This shows the general robustness of our metric when used to compare methods for different applications.
>
>
> Inclusion of the Adjusted Rand Index (ARI):
> Based on your suggestion, we have added evaluation with respect to the ARI in the revision, including discussion in the results section (Section 5 text, Fig 6).  We found that the ARI tends to mediate the effect of changing the number of clusters compared to NMI as you’d suggested. However, it suffers from the same problems as NMI in evaluating temporal clusterings, without the benefit of having constituent criteria that can be analyzed and interpreted.
>
>
> Difficulty of analyzing Munkres:
> We have improved the clarity of the argument against the difficulty of using the Munkres metric. The Munkres method has two main issues: (i) Since it relies on computing a matching between ground-truth labels and clusters, the score is agnostic to changes in clusters that are not matched with any ground-truth label. This “problem of matching” was pointed out in Rosenberg and Hirschberg (2007) and Meila (2007) for standard clustering settings. (ii) A contingency matrix is fed to the Munkres method for computing the optimal correspondences, ignoring temporal structure.
>
>
> Segment length generation:
> We have updated our discussion to clarify that the generative process we propose (sample m Categoricals from the prior and sort them) is exactly equivalent to generating segment lengths from a Multinomial distribution (over m draws) with a Dirichlet prior. However, representing the process in the way we have written it improved inference efficiency with Gibbs sampling -- resampling the segment lengths requires only computing the likelihood of data points at segment boundaries, which is independent of the length of the time-series and far more efficient (we also discuss this in the appendix).

---

> > ### Author Response · Authors · 2018-11-27
> > **Author Response (continued)**
> >
> > Hyperparameters in experimental evaluation:
> > We have added further experimentation to show Prism’s sensitivity to the number of segments (s) and clusters (K). We found that Prism’s performance is relatively insensitive to the number of segments as long as it is greater than the number in ground-truth, suggesting one can set it to a large value. Prism’s performance is also stable across a wide range of K.

---

### Official Review · AnonReviewer3 · 2018-11-03
**some good ideas, but performance metric isn't sufficiently compared or validated, model contributions aren't enough**

**Rating:** 5
**Confidence:** 2

**Review:**

This is a hybrid paper, making contributions on two related fronts:
1. the paper proposes a performance metric for sequence labeling, capturing salient qualities missed by other metrics, and
2. the paper also proposes a new sequence labeling method based on inference in a hierarchical Bayesian model, focused on simultaneously labeling multiple sequences that have the same underlying procedure but with varying segment lengths.


This paper is not a great topic fit for ICLR: it's primarily about a hand-designed performance metric for sequence labeling and a hierarchical Bayesian model with Gaussian observations and fit with Gibbs sampling in a full-batch setting. The ICLR 2019 reviewer guidelines suggest "Ask yourself: will a substantial fraction of ICLR attendees be interested in reading this paper?" and based on my understanding of the ICLR audience I suspect not. Based on looking at past ICLR proceedings, this paper's topic and collection of techniques is not in the ICLR mainstream (though it's not totally unrelated). The authors could convince me that I'm mistaken by pointing out closely related ICLR papers (e.g. with a similar mix of techniques in their methods, or similarly proposing a hand-designed performance metric); as far as I can tell, none of the papers cited in the references are from ICLR, but rather from e.g. NIPS, AISTATS, and IEEE TPAMI, which I believe would be better fits for this kind of work.

One way to make this work more relevant to the ICLR audience would be to add feature learning (especially based on neural network architectures). That might also entail additional technical contributions, like how to fit models like these in the minibatch setting (where the current Gibbs sampling method might not apply).


On the proposed performance metric, the discussion of existing metrics as they apply to the example in Fig 3 was really helpful. (I assume, but didn't check, that the authors' characterization of the published performance metrics is accurate, e.g. "no traditional clustering criteria can distinguish C_2 from C_3".) The proposed metric seems to help.

But it's a bit complicated, with several free design decisions involved (e.g. choosing the scoring function \mathcal{H} in Sec 3.1, the choice of conditional entropy H in Sec 3.2, the choice of \beta in Sec 3.3, the choice of the specific algebraic forms of RSS, LASS, SSS, and TSS). Certainly the proposed metrics incorporate the kind of information that the authors argue can be important, but the design details of how that information is summarized into a single number aren't really explored or weighed against alternative designs choices.

If a primary aim of this paper is to propose a new performance metric, and presumably to have it catch on with the rest of the field, then the contribution would be much greater if the design space was clearly articulated, alternatives were considered, and multiple proposals were validated. Validation could be done with human labelers ranking the intuitive 'goodness' of labeling results (and then compared to rankings derived from the proposed performance metrics), and with comparing how the metrics correlate with performance on various downstream tasks.

Another idea is to take advantage of a better segmentation performance metric and use it to automatically tune the hyperparameters of the sequence labeling methods considered in the experiments section. (IIUC hyperparameters were set by hand in the experiments.). That would make for more interesting experiments that give a more comprehensive summary of how these techniques can compare.

However, as it stands, while the performance metric itself may have merit, in this paper it is not sufficiently well validated or compared to alternatives.


On the hierarchical Bayesian model, the current model design andinference algorithm are okay but don't constitute major technical contributions. I was surprised by some model details: for example, in "Modeling the procedure" of Sec 4.1, it would be much more satisfying to generate the (p_1, ..., p_s) sequence from an HMM instead of sampling the elements of the sequence independently, dropping any chance to learn transition structure as part of the Bayesian inference procedure. More importantly, it wasn't made clear if 'self-transitions' where p_s = p_{s+1} were ruled out, though such transitions might confuse the model's semantics. As another example, in "Modeling the realizations in each time-series" of Sec 4.1, the procedure based on iid sampling and sorting seems unnatural, and might make inference more complex. Why not just sample the durations directly (rather than indirectly defining them via sorting independently-generated indices)? If there's a good reason, it should probably be discussed (e.g. maybe parameterizing the durations directly would make it easier to express prior distributions over *absolute* segment lengths, but harder to express distributions over *relative* segment lengths?). Finally, the restriction to conditionally iid Gaussian observations was disappointing.

The experimental results were solid on the task for which the model's extra assumptions paid off, but that's a niche comparison.

One suggestion on the baseline front: you can tie multiple HMMs to have the same procedure (i.e. the same state sequences not counting repeats) by fixing the number of states to be s (the length of the procedure sequence) and fixing the transition matrices to have an upper-bidiagonal support structure. A similar construction can be used for HSMMs. I think a natural Gibbs sampling procedure would emerge. This approach is probably written down in the HMM literature (it seems every conceivable HMM variant has been studied!) but I don't have a reference for it.


Overall, this paper needs more work.


Minor suggestions:
- maybe refer to "segment structure" (e.g. in Sec 3), as "changepoint structure" (and consider looking into changepoint performance metrics if you haven't already)
- if you used code from other authors in your baselines, it would be good to cite that code (e.g. GitHub links)

---

> ### Author Response · Authors · 2018-11-27
> **Author Response (metrics)**
>
> Thank you for your detailed comments! Here’s a response to the concerns that you raised,
>
> Response on metrics:
>
> Indeed, we do view our primary contribution as providing a new performance metric for better assessing the quality of extracting latent structure in temporal sequences. We appreciate the suggestion to (i) more clearly articulate the design space; (ii) discuss alternative formulations and (iii) validate other proposals. We have updated our paper in Sections 3 & 5 as well as Figure 6 to address this and we briefly summarize this here.
>
> We began from the stance that something seemed to be missing in existing evaluation criteria that is important to capture in temporal data -- segment and repeated structure. We wanted to draw on the widely used criteria that have been designed for evaluating clusterings (Rosenberg and Hirschberg 2007) and consider them in the temporal setting. In doing so we encountered a multi-objective problem, in how to weigh the new criteria that we designed (RSS and SSS). Our solution to introduce a tradeoff parameter (\beta) follows the approach laid out in the key paper on (non-temporal) clustering evaluation by Rosenberg and Hirschberg (2007) that introduces the V-measure for clustering evaluation. That paper includes a tradeoff parameter between completeness and homogeneity (the constituent criteria for V-measure) with the harmonic mean (\beta=1) kept as the default, allowing them to “prioritize one criterion over another, depending on the clustering task and goals.”
>
> Based on your and other reviewers’ excellent feedback to further explore how these settings impact the resulting metric, we have also introduced a new sensitivity analysis for the tradeoff parameter. While determining the right value of \beta for evaluation is a function of the problem and end-goal (e.g. repeated structure may have no importance in changepoint segmentation, and should be disregarded), we show that \beta=1 is a problem-agnostic compromise that works well in practice. To this end, our sensitivity analysis answers the following question: suppose \beta’ =/= 1.0 is the right value of the tradeoff parameter for a particular problem -- how similar does the metric perform (in terms of the ranking of methods) by using \beta=1.0? We find, quite naturally, that at extreme values of \beta (0 or \infty) the methods may be ranked quite differently than by \beta=1.0. However, a large range of \beta values can be approximated well by using \beta=1.0 (Figure 6i in the revised paper). This shows the general robustness of our metric when used to compare methods for different applications.
>
> We completely agree that it would be interesting to conduct a study in which human judgments are compared to the evaluation criteria to validate them qualitatively. Interestingly, prior work in the area such as Rosenberg and Hirschberg (2007), Meila (2007), Dom (2001) also did not conduct user studies but instead justified their metric through examples and direct reasoning about how it fulfils the desiderata laid out; we followed this in our work, with the additional inclusion of a large-scale comparison of methods on real-world datasets, as well as validation of the tradeoff parameter that we introduced.
>
> We believe those working on time-series data will benefit from having access to the tailored evaluation criteria we have introduced. These evaluation criteria identify and target specific characteristics of the temporal clustering setting, something that has not been done systematically in the past.

---

> > ### Author Response · Authors · 2018-11-27
> > **Author Response (method)**
> >
> > We would like to highlight that we see our new evaluation criteria as our primary contribution, and here we are of course happy to clarify questions about the algorithm we introduced (Prism), which provides a small concrete improvement in trying to model procedural structure.
> >
> >
> > Segment length generation:
> > We have updated our discussion in Section 4 to clarify that the generative process we propose (sample m Categoricals from the prior and sort them) is exactly equivalent to generating segment lengths from a Multinomial distribution (over m draws) with a Dirichlet prior. However, representing the process in the way we have written it improved inference efficiency with Gibbs sampling -- resampling the segment lengths requires only computing the likelihood of data points at segment boundaries, which is independent of the length of the time-series and far more efficient (we also discuss this in the appendix).
> >
> >
> > Baseline suggestion:
> > The baseline suggested is interesting -- however, our concern is that it cannot represent repeated structure since the bidiagonal structure only allows forward transitions. As a key part of our proposed metrics is to be able to capture repeated structure, it would be somewhat impoverished in comparison to our method and the HMM models we compare to.
> >
> >
> > Generate procedure as HMM:
> > The decision to generate each step in the procedure (p_1, …, p_s) independently was a conscious design choice to improve the model’s ability to recover non-Markov segmentations. Non-Markov processes can be made Markov by expanding the state (to include the history) but this can both increase the amount of data needed to fit a good model (since there are now more states) and make it harder to identify repeated structure. Alternatively, fitting a non-Markov procedure using a Markov process can result in learning a highly stochastic transition model that does not provide a good fit to the data. We explored both of these possibilities and found as expected that they did not do well in the simulations we considered, though we completely agree that we could also use a Markov process to generate the procedure, if it is present. We have included a small simulation study in the appendix that highlights this point.
> >
> >
> > Presence of self-transitions:
> > Our model allows self-transitions which can lead to adjacent segments being assigned the same label (effectively causing them to be condensed into a single segment). A version of our model that rules out self-transitions performed similarly, so we have not included that in the paper to avoid confusion, since inference for that model is far more complex and requires the introduction of auxiliary variables in the model.
> >
> >
> > Mini-batch learning/neural net observations:
> > While we expect that these extensions will allow us to scale directly to high-dimensional data and improve performance, our focus was to establish the need for learning procedural abstractions. Procedural tasks are extremely common, and our experiments show the benefit of baking in structural assumptions into the data-generating process. We used the same observational model for all compared methods to disentangle this benefit. Recent related work such as Johnson et al. (2016) (“Composing graphical models with neural networks for structured representations and fast inference”) could provide a way of combining the kind of structured model we have described with neural net observations. Another alternative would be to design a variational autoencoder using the Gumbel-Softmax trick to represent the discrete variables. However, these are non-trivial extensions that require careful thought so we defer them to future work.
> >
> >
> > Hyperparameters in experimental evaluation:
> > We have added further experimentation to show Prism’s sensitivity to the number of segments (s) and clusters (K) in Section 6 and Figure 7. We found that Prism’s performance is relatively insensitive to the number of segments as long as it is greater than the number in ground-truth, suggesting one can set it to a large value. Prism’s performance is also stable across a wide range of K.

---

> > > ### Author Response · Authors · 2018-11-27
> > > **Author Response (clarifications)**
> > >
> > > We also discuss additional clarifications to questions raised by you,
> > >
> > > Choosing \mathcal{H} (S3.1):
> > > While the space of alternative choices is enormous, the main desiderata is that the chosen scoring function should look for overlapping sequences of tokens in the two segments, so common substring/subsequence style functions are most suitable. Using the heaviest common substring allows us to take into account the relative length of the matched token sequence.
> > >
> > >
> > > Clarification for H (S3.2):
> > > This follows from the definition of conditional entropy, as laid out in prior work such as Rosenberg & Hirschberg (2007), Meila (2007) and Dom (2001), who adopt these definitions for the standard clustering setting.
> > >
> > >
> > > Algebraic forms of criteria:
> > > The choice of algebraic form is to represent them as normalized mutual information criteria. The metric that is canonically called NMI is one instance of a family of such criteria. The criteria we derived are part of this family and can be rewritten as a mutual information term divided by some normalization. For instance,
> > >
> > > LASS 	= 1 - \frac{H(A|B) + H(B|A)} {H(A) + H(B)}
> > >                 = \frac{H(A) + H(B) - H(A|B) - H(B|A)} {H(A) + H(B)}
> > >                 = \frac{2 * I(A;B)} {H(A) + H(B)}
> > >
> > > This relates them to the large body of previous work in clustering evaluation using information-theoretic criteria (see Table 2 in Vinh, Epps and Bailey (2010) for a review).
> > >
> > > -------------------------------------------------------
> > > We would like to conclude by thanking you for the helpful suggestions. We hope that our response addresses the concerns raised by you and that you will reconsider your assessment of our work.

---

> ### Comment · AnonReviewer3 · 2018-12-15
> **modestly revising up my review**
>
> The author responses to my review were thorough and compelling. The revisions made the paper stronger.
>
> One of my main complaints about the paper was that it might not be a good subject fit for ICLR. That the other reviewers did not raise the same objection (indeed thought the opposite: "This work is appropriate for ICLR."), and gave positive reviews, leads me to believe I could be wrong about the subject fit. That is, my confidence in my evaluation is now lower.
>
> I still believe the contribution in this manuscript would be much stronger if (1) it contained user studies that showed the proposed metric corresponds to some human perception of the goodness of segmentation or (2) it showed that improvements on the metric correlated with some kind of downstream task performance. Without a compelling demonstration of strengths like these, it seems much less likely that the metric or the proposed method will impact others' future work.
>
> I'll revise my review score up to be on the negative side of neutral, and revise down my confidence. That way I expect my review wouldn't be enough to sink the submission if another reviewer wants to champion it.

---

### Official Review · AnonReviewer2 · 2018-11-05
**Learning procedural abstractions and evaluating discrete latent temporal structure**

**Rating:** 6
**Confidence:** 3

**Review:**

In "Learning procedural abstractions and evaluating discrete latent temporal structure" the authors develop a hierarchical Bayesian model for patterns across time in video data. They also introduce new metrics for understanding structure in time series (completeness and homogeneity). This work is appropriate for ICLR. They provide some applications to robotics, suggesting that this could be used to teach robots to act in environments by learning from videos.

This manuscript paid quite close attention to quality of segmentation, in which actions in videos are decomposed into component parts. It is quite hard to determine groundtruth in such situations and many metrics abound, and so a thorough discussion and comparison of metrics is useful.

The state of the art for Bayesian hierarchical models for segmentation is Fox et al., which is referenced heavily by this work (including the use of test data prepared in Fox et al.) I wonder why the authors drop the Bayesian nonparametric nature of the hierarchy in the section "Modeling realizations in each time-series" (i.e., for Fox et al., the first unnumbered equation in this section would have had arbitrary s).

I found that the experiments were quite thorough, with many methods and metrics compared. However, I found the details of the model to be quite sparse, for example it's unclear how Figure 5 is that much different from Fox et al. But, overall I found this to be a strong paper.

---

> ### Author Response · Authors · 2018-11-27
> **Author Response**
>
> Thank you for your encouraging comments! Here’s a response to the concerns that you raised,
>
> Incorporating nonparametric priors:
>
> We completely agree incorporating a nonparametric prior would be an interesting extension to our approach.  We chose not to incorporate this in our current work for several reasons:
>
> (i) Our contribution was to establish the benefit of incorporating additional assumptions about the underlying procedure with the ability to flexibly learn non-Markov procedures. Thus, keeping the model as simple as possible allows us to isolate the difference that our model makes compared to existing methods that typically make more restrictive, Markovian assumptions. Using a nonparametric prior adds an additional confounder, complicating our ability to understand whether the benefit is caused by the prior, or by the modeling assumptions used. We anticipated incorporating a nonparametric prior would offer no additional benefit beyond the ability to flexibly set some quantities based on data. Fox et al.’s central contribution was to describe new inference methods for such priors, while the focus of our work is different -- understanding where existing modeling fall short in modeling procedural data, and addressing them.
>
> (ii) Even without a nonparametric prior, Prism has the ability to ‘skip’ steps in the procedure. This can be realized since the model is able to set segment lengths to be 0 for some steps in the procedure. Thus, we can achieve at least some of the flexibility afforded by the nonparametric prior by setting the number of segments to be large. We have added further experimentation in the revision to show Prism’s sensitivity to the number of segments (s) -- see Figure 7. We found that Prism’s performance is relatively insensitive to the number of segments as long as it is larger than the number in ground-truth.
>
> Distinction from Fox et al:
>
> A related concern that was pointed out is how Fig. 5 is distinct from the work of Fox et al. Fox et al. primarily target recovering a faithful generative model for the sequences. In contrast we focus on identifying the latent structure in the given sequences. In particular, for a procedure identification setting, we describe how sharing a common procedure (not done in Fox et al.) and separating only the realizations (also different from Fox et al., which assumes a Markov stochastic transition matrix) can improve our ability to recover the latent segmentation. Technically these distinctions lie in how we model the data-generating process, specifically the local assignments of each data-point to a latent discrete cluster label. Fox et al. are concerned with the specification of and inference for, nonparametric priors that can be used with autoregressive generative HMM/SLDS models, in contrast to our work. We believe that identifying latent segmentation structure alone (even sans a generative model) is often of useful value, such as for the important application of activity understanding, or potentially for identifying building blocks in imitation learning.

---

### Meta-Review · Area_Chair1 · 2018-12-14
**Meta-Review for Learning Procedural Abstractions**

**Confidence:** 3
**Recommendation:** Accept (Poster)

**Metareview:**

While the reviews of this paper were somewhat mixed (7,6,4), I ended up favoring acceptance because of the thorough author responses, and the novelty of what is being examined.

The reviewer with a score of 4, argues that this work is not a good fit for iclr, but, although tailoring new metrics may not be a common area that is explored, I don't believe that it's outside the range of iclr's interest, and therefore also more unique.